# CAUSAL CONTEXTUAL BANDITS WITH TARGETED INTERVENTIONS

**Chandrasekar Subramanian[1, 2], Balaraman Ravindran[1, 2]**
[1] Robert Bosch Centre for Data Science and Artificial Intelligence
[2] Department of Computer Science and Engineering
Indian Institute of Technology Madras, Chennai, India
`sekarnet@gmail.com, ravi@cse.iitm.ac.in`

## ABSTRACT

We study a contextual bandit setting where the learning agent has the ability to perform interventions on targeted subsets of the population, apart from possessing qualitative causal side-information. This novel formalism captures intricacies in real-world scenarios such as software product experimentation where targeted experiments can be conducted. However, this fundamentally changes the set of options that the agent has, compared to standard contextual bandit settings, necessitating new techniques. This is also the first work that integrates causal side-information in a contextual bandit setting, where the agent aims to learn a policy that maps contexts to arms (as opposed to just identifying one best arm). We propose a new algorithm, which we show empirically performs better than baselines on experiments that use purely synthetic data and on real world-inspired experiments. We also prove a bound on regret that theoretically guards performance.

## 1 INTRODUCTION

Contextual bandits have been used as natural frameworks to model interactive decision making scenarios such as recommendation systems (Liu et al., 2018), marketing campaign allocation (Sawant et al., 2018) and more (Bouneffouf & Rish, 2019). In this framework, the learning agent repeatedly interacts with an *environment* with the aim of learning a near-optimal decision *policy* that maps a context space to a set of actions (also referred to as arms or interventions[1]). In the standard stochastic variant, in each round of interaction, the agent observes a context from the environment and, once the agent chooses an action, the environment returns a sampled reward that is a function of the current context and chosen action (Lattimore & Szepesvári, 2020). The agent's objective is to minimize some meaningful measure of closeness of the policy to optimality (see Lattimore & Szepesvári (2020) for some standard definitions).

One of the key issues in the wide application of contextual bandits (and reinforcement learning, in general (Dulac-Arnold et al., 2021)) is their need for a large number of samples that are costly to actualize in practice. For example, each arm might correspond to performing a product experiment on users or to conducting a specific medical intervention. However, we show that there are nuances in real-world situations which – though not fully captured by the standard formulations of contextual bandits – if modeled and leveraged, allow us to build methods that can improve the rate at which good policies can be identified.

**A motivating example**   Consider a sales-assistance software agent that is learning to suggest a campaign that is optimal for a given sales lead (defined by a set of features). With software products, there is often an opportunity to conduct *experiments* with variants of the software on defined *subsets* of the population of users (each specified by a set of characteristics, or context variables) to learn about resulting metrics; see Google (2021) for an example. We can think of these experiments as constituting a *training phase*. More specifically, the agent, for example, can conduct a campaign

---

[1]We use the term *intervention* here because actions or arms in a (contextual) bandit setting can be interpreted as Pearl $do()$ interventions on a causal model (Zhang & Bareinboim, 2017; Lattimore et al., 2016). See Pearl (2009; 2019) for more discussion on the $do()$ operation.

(i.e., the intervention) on a specific subset of users (for example, the context value `os=iOS` could define a subset) and observe the outcome. Thus, instead of necessarily performing an intervention on a randomly sampled user coming from the natural population distribution, the agent can instead choose to target the intervention on a randomly selected user *with specific characteristics* (as defined by an assignment of context variable values). This type of interventions, which we call *targeted interventions*, fundamentally changes the set of options that the agent has in every training round.

In addition, the training phase often happens in a more lenient environment where the agent might have access to auxiliary context variables (such as `IT-spend`) that are unavailable in the evaluation phase. Further, there is also causal side-information sometimes available. For instance, we might know that `emailsubject` causes `openemail`, and not the other way around. Encoding this qualitative side-information as a causal graph can help the agent make use of this structure. Lattimore et al. (2016) and Yabe et al. (2018) demonstrate this in the best-arm identification case. After training, there is an evaluation phase (e.g., when the agent is deployed), where the agent observes the main context variables and decides an action; its regret performance is measured at this point.

**Our framework** Our framework captures the above intricacies (which are not restricted to software product experimentation). The formal treatment is in Section 2; we present an overview here. The agent has $T$ rounds of interaction that act as the training phase, followed by a $(T + 1)$'th round on which it is evaluated. In each of the $T$ training rounds, it has the choice to perform either a targeted intervention or a standard interaction. Further, in addition to the main context variables, the agent can observe a (possibly empty) set of additional auxiliary context variables during training; these auxiliary context variables are not observable during evaluation. *All* variables are made available to the agent at the end of every training round (this is similar to Lattimore et al. (2016)). The agent also has access to a causal graph that models the qualitative causal relationships between the variables; there are very few assumptions made on this graph (discussed in Section 2.1). The causal graph allows a factorized representation of the joint distribution of the variables, and as a result, enables *information leakage* – i.e., updating beliefs about several interventions after every single intervention. Also, importantly, we allow context variables to be categorical, and therefore the usual assumptions that enable generalization of the learned policy across contexts, such as linearity (e.g., in Dimakopoulou et al. (2019)), become invalid.

The agent's problem is one of sample allocation – how to allocate samples across the $T$ training rounds so as to learn a policy that minimizes regret in the $(T + 1)$'th round that represents the evaluation phase. In the evaluation phase, the agent observes the main context variables from the environment, against which it chooses an action and receives a reward – much like in a standard contextual bandit setting. Since targeted interventions are restricted to the training phase, it introduces a difference between the training and evaluation phases. The agent's challenge, therefore, is to learn policies that minimize regret in the evaluation phase using samples it collects in the training phase (from a combination of standard interactions and targeted interventions). The algorithm we propose utilizes a novel entropy-*like* measure called Unc that guides this sample allocation in a way that also exploits the information leakage.

## 1.1 CONTRIBUTIONS

In this paper, we formalize this modified setting, which we call "causal contextual bandits with targeted interventions", provide a novel algorithm and show both theoretical and experimental results. Specifically, our contributions are:

- We formalize the more nuanced contextual bandit setting described above (Sec. 2). This is the first work that we know of that formulates a *contextual* bandit setting with causal side-information. This is also the first paper we are aware of that introduces targeted interventions in a contextual bandit setting.

- We propose a new algorithm based on minimizing a novel entropy-*like* measure (Sec. 3.1)

- We prove a bound on its regret, providing a theoretical guard on its performance (Sec. 3.2)

- We show results of experiments that use purely synthetic data (Sec. 4.1). The results demonstrate that our algorithm performs better than baselines.

- We also run experiments that are inspired by proprietary data from Freshworks Inc. (Sec. 4.2), providing evidence of our algorithm's performance in more realistic scenarios as well.

Our motivation comes from real world settings where experiments are costly; therefore, we are interested in empirical behavior when the training budget $T$ is relatively small.

## 1.2 RELATED WORK

Causal bandits have been studied in literature recently (Lattimore et al., 2016; Sen et al., 2017; Yabe et al., 2018; Lu et al., 2020) and they leverage causal side-information to transfer knowledge across interventions. They, however, have been studied only in a best arm identification setting, with each arm modeled as an intervention on a set of variables; the objective is to learn a single best arm. Our setting differs significantly since the agent attempts to learn a *policy* (by which we mean a mapping from contexts to arms) instead of a single best arm. Therefore, while the agent can perform targeted interventions that specify both the context and action, it is still attempting to learn an optimal context-action mapping.

The contextual bandit literature is well-studied (see Zhou (2016) and Lattimore & Szepesvári (2020) for surveys), and has taken various approaches to enable knowledge transfer across contexts. For example, Dimakopoulou et al. (2019) assume expected reward is a linear function of context, while Agarwal et al. (2014) make assumptions about the existence of a certain oracle. However, there has not been work that has looked at contextual bandits which utilize causal side-information as a way to transfer knowledge across contexts, or considered targeted interventions in the space of options for the agent. Our method utilizes the information leakage that the causal graph provides to not just learn about other interventions after every intervention, but also to be smarter about the choice of targeted interventions to conduct.

Another related area is active learning (Settles, 2009). In active learning, a supervised learning agent gets the option to query an oracle actively to obtain the true labels for certain data points. However, it is in the supervised learning setting where the agent receives true label feedback, whereas in our setting the agent only receives bandit feedback (that is, only for the action that was taken). Nevertheless, our work can be thought of as infusing some elements of the active learning problem into the contextual bandits setting by providing the agent the ability to perform targeted interventions. There has been some work investigating the intersection of causality and bandits with the aim of transfer learning. Zhang & Bareinboim (2017) study transfer of knowledge from offline data in the presence of unobserved confounders, in a non-contextual setting. As can be seen, this differs significantly from our setting.

## 2 FORMALISM

We assume that the underlying environment is modeled as a *causal model* $\mathcal{M}$, which is defined by a directed acyclic graph $\mathcal{G}$ over all variables – i.e., the variable to be intervened ($X$), the reward variable ($Y$), and the set of context variables ($\mathcal{C}$) – and a joint probability distribution $\mathbb{P}$ that factorizes over $\mathcal{G}$ (Pearl, 2009; Koller & Friedman, 2009). $\mathcal{C}$ is partitioned into the set of main context variables ($\mathcal{C}^{tar}$) and the set of auxiliary context variables ($\mathcal{C}^{other}$). $\mathcal{G}$ is sometimes called the *causal graph* or *causal diagram* of $\mathcal{M}$. Each variable takes on a finite, known set of values; note that this is quite generic, and allows for categorical variables. An *intervention* $do(X = x)$ on $\mathcal{M}$ involves removing all arrows from the parents of $X$ into $X$, and setting $X = x$ (Pearl, 2009). The agent has access only to $\mathcal{G}$ and not to $\mathcal{M}$; therefore, the agent is not given any knowledge *a priori* about the underlying conditional probability distributions (CPDs) of the variables.

We specify a targeted intervention $X = x$ conditioned on context $\mathcal{C}^{tar} = \mathbf{c}^{tar}$ succinctly by $(x, \mathbf{c}^{tar})$. While standard interventions (Pearl, 2009) result in a distribution of the form $\mathbb{P}(. \mid do(x))$, targeted interventions result in $\mathbb{P}(. \mid do(x), \mathcal{C}^{tar} = \mathbf{c}^{tar})$. Table 1 summarizes the key notation used in this paper.

The agent is allowed $T$ training rounds of interaction with the environment, at the end of which it aims to have learned a policy $\hat{\phi} : \mathsf{val}(\mathcal{C}^{tar}) \to \mathsf{val}(X)$. Specifically, in each of the $T$ training rounds, the agent can choose to either

- *(Standard interaction)* Observe context $\mathbf{c}^{tar} \sim \mathbb{P}(\mathcal{C}^{tar})$, choose intervention $x$, and observe $(y, \mathbf{c}^{other}) \sim \mathbb{P}(Y, \mathcal{C}^{other} \mid do(x), \mathbf{c}^{tar})$, *(or)*

Table 1: Summary of key notation

| Symbol/Notation | Meaning |
|---|---|
| $X$ | variable that will be intervened upon |
| $Y$ | reward variable |
| $\mathcal{C}^{tar}, \mathcal{C}^{other}$ | set of main context variables and set of auxiliary context variables, respectively; so the set of all context variables is $\mathcal{C} = \mathcal{C}^{tar} \cup \mathcal{C}^{other} = \{..., C_i, ...\}$. |
| Capital letters | a random variable; e.g., $C_1$ or $X$ |
| Small letters | a random variable's value; e.g., $c_1$ or $x$ |
| Small bold font | a vector of random variable values; e.g., $\mathbf{c}^{tar}$ denotes a specific choice of values taken by variables in $\mathcal{C}^{tar}$ |
| $(x, \mathbf{c}^{tar})$ | specifies targeted intervention $X = x$ on subset defined by $\mathcal{C}^{tar} = \mathbf{c}^{tar}$ |
| $\hat{\mathbb{P}}, \hat{\mathbb{E}}$ | estimate of distribution $\mathbb{P}$ and expectation $\mathbb{E}$ based on current beliefs |
| $\theta_{V\|\mathbf{pa}_V}$ | beliefs (vector of Dirichlet parameters) about parameters of $\mathbb{P}(V\|\mathbf{pa}_V)$; $\theta_{V\|\mathbf{pa}_V}^{(v)}$ is the entry corresponding to $V = v$. |
| $\mathsf{val}(V), \mathsf{val}(\mathcal{V})$ | set of values taken by the variable $V$, and set of variables $\mathcal{V}$, respectively. |
| $N_V, N_\mathcal{V}$ | $= \|\mathsf{val}(V)\|, \|\mathsf{val}(\mathcal{V})\|$ |
| $\mathbf{pa}_V$ | value of variables in $PA_V$, the parents of $V$. For $Y$, we let $PA_Y$ denote its parents *excluding* $X$ to keep proof easier to read. |
| $T$ | number of training rounds |
| $\alpha$ | fraction of rounds in Phase 1 of training |
| $\mathbf{a}\langle\mathcal{B}\rangle$ | if $\mathbf{a}$ is an assignment of values to $\mathcal{A}$, then $\mathbf{a}\langle\mathcal{B}\rangle$ is assignment of those values to respective variables $\mathcal{B} \subseteq \mathcal{A}$. |
| Set operations | for readability, we use them on vectors as well, where there is no ambiguity. |

- *(Targeted intervention)* Choose intervention $x$ and a target subset given by context values $\mathcal{C}^{tar} = \mathbf{c}^{tar}$, and observe $(y, \mathbf{c}^{other}) \sim \mathbb{P}(Y, \mathcal{C}^{other} \mid do(x), \mathbf{c}^{tar})$

where, when there is no ambiguity, we use $\mathbb{P}$ interchangeably to mean either the joint distribution or a marginal distribution. Note that in both modes of interaction above, the intervention is only on $X$; they differ, however, in whether the agent *observes* the context values from the environment or whether it *chooses* the context values on which to condition the intervention.

After training, the agent is evaluated in the $(T + 1)$'th round. Here, the agent is presented a query context $\mathbf{c}^{tar} \sim \mathbb{P}(\mathcal{C}^{tar})$ to which it responds with an action $x = \hat{\phi}(\mathbf{c}^{tar})$ using the learned policy, and receives a reward $y \sim \mathbb{P}(Y \mid do(x), \mathbf{c}^{tar})$ from the environment.

The objective of the agent is to learn a policy $\hat{\phi}$ that minimizes *simple-regret*, which is defined as:

$$\text{Regret} \triangleq \sum_{\mathbf{c}^{tar}} [\mu^*_{\mathbf{c}^{tar}} - \hat{\mu}_{\mathbf{c}^{tar}}] \cdot \mathbb{P}(\mathbf{c}^{tar}) = \sum_{\mathbf{c}^{tar}} \text{Regret}(\mathbf{c}^{tar}) \cdot \mathbb{P}(\mathbf{c}^{tar})$$

where $\phi^*$ is an optimal policy, $\mu^*_{\mathbf{c}^{tar}} \triangleq \mathbb{E}[Y|do(\phi^*(\mathbf{c}^{tar})), \mathbf{c}^{tar}]$ and $\hat{\mu}_{\mathbf{c}^{tar}} \triangleq \mathbb{E}[Y|do(\hat{\phi}(\mathbf{c}^{tar})), \mathbf{c}^{tar}]$.

## 2.1 ASSUMPTIONS

We are interested in DAGs $\mathcal{G}$ where there is an edge $X \to Y$, and no other directed path between $X$ and $Y$; that is, $X$ directly affects $Y$; there is no context variable *caused by* $X$. This models the natural situation we encounter in settings such as a recommendation system where click-through rate ($Y$) of a user is directly affected by the user's features (the context) and the recommendation ($X$); the recommendation does not cause any contexts, so doesn't have a path to $Y$ through them. We assume that there are no unobserved confounders (UCs); but purely interactive bandit settings, such as ours, are robust to *direct* UCs between $X$ and $Y$ since actions can be interpreted as sampling from the $\mathbb{P}(.|do(x))$ distribution (Bareinboim et al., 2015; Guo et al., 2020). We make an additional assumption to simplify the factorization in Section 3.2: (**A1**) $\{C$ confounds $C' \in \mathcal{C}^{tar}$ and $Y\} \implies C \in \mathcal{C}^{tar}$. A *sufficient* condition for (**A1**) to be true is if $\mathcal{C}^{tar}$ is ancestral (i.e., $\mathcal{C}^{tar}$ contains all its ancestors). This assumption does not preclude UCs (for example, the graph used in the synthetic experiments in Section 4.1 could have a UC between $X$ and $Y$), and can be relaxed in the future.

## 3 SOLUTION APPROACH

### 3.1 ALGORITHM (UNC_CCB)

---

**Algorithm 1a:** Training phase of Unc_CCB

---

**Data:** Graph $\mathcal{G}$; fraction $\alpha$ of Phase 1 rounds.
**Initialization:** For all $V \in \mathcal{C} \cup \{Y\}$, set $\theta_{V|\mathbf{pa}_V} = (1, ..., 1)$, for all values of $\mathbf{pa}_V$.

---

1 *(Phase 1)* **for** $t = 1 \ldots \lceil \alpha T \rceil$ **do**
2     Observe context $\mathbf{c}^{tar} \sim \mathbb{P}(\mathcal{C}^{tar})$
3     Choose $x \in \mathsf{val}(X)$ uniformly at random
4     Observe $(y, \mathbf{c}^{other}) \sim \mathbb{P}(Y, \mathcal{C}^{other} \mid do(x), \mathbf{c}^{tar})$
5     **for** $V$ *in* $\mathcal{C}$ **do**
6        **updateBeliefs**$(V, \mathbf{c}\langle PA_V \rangle, \mathbf{c}\langle V \rangle)$   `//denote `$\mathbf{c}^{tar} \cup \mathbf{c}^{other}$` by `$\mathbf{c}$
7     **end**
8     **updateBeliefs**$(Y, (\mathbf{c}\langle PA_Y \rangle, x), \mathbf{c}\langle Y \rangle)$
9 **end**
10 *(Phase 2)* **for** $t = \lceil \alpha T \rceil + 1, \ldots, T$ **do**
11     Choose $x, \mathbf{c}^{tar}$ as:

$$\underset{x \in \mathsf{val}(X),\ \mathbf{c}^{tar} \in \mathsf{val}(\mathcal{C}^{tar})}{\arg\min} \left\{ \sum_{x' \in \mathsf{val}(X),\ \mathbf{c}^{tar\prime} \in \mathsf{val}(\mathcal{C}^{tar})} \mathsf{Unc}\left( \mathbb{E}[Y|do(x'), \mathbf{c}^{tar\prime}] \mid x, \mathbf{c}^{tar} \right) \right\}$$

12     Do targeted intervention $(x, \mathbf{c}^{tar})$ and observe $(y, \mathbf{c}^{other}) \sim \mathbb{P}(Y, \mathcal{C}^{other}|do(x), \mathbf{c}^{tar})$
13     **for** $V$ *in* $\mathcal{C}^{other}$ **do**
14        **updateBeliefs**$(V, \mathbf{c}\langle PA_V \rangle, \mathbf{c}\langle V \rangle)$
15     **end**
16     **updateBeliefs**$(Y, (x, \mathbf{c}\langle PA_Y \rangle), \mathbf{c}\langle Y \rangle)$
17 **end**

**Result:** Final set of beliefs for all $V, \mathbf{pa}_V$: $\left\{ ..., \theta_{V|\mathbf{pa}_V}, ... \right\}$

---

18 *Procedure* **updateBeliefs**$(V, \mathbf{pa}_V, v)$
19     $\theta_{V|\mathbf{pa}_V}^{(v)} \leftarrow \theta_{V|\mathbf{pa}_V}^{(v)} + 1$

---

**Algorithm 1b:** Evaluation phase of Unc_CCB

---

**Data:** Graph $\mathcal{G}$, learned beliefs $\left\{ ..., \theta_{V|\mathbf{pa}_V}, ... \right\}$, user context to be decided $\mathbf{c}^{tar}$

---

1 **for** *every* $V, \mathbf{pa}_V$ **do**
2     **for** $v \in \mathsf{val}(V)$ **do**
3        Set $\hat{\mathbb{P}}(V = v | \mathbf{pa}_V) = \dfrac{\theta_{V|\mathbf{pa}_V}^{(v)}}{\sum_{v'} \theta_{V|\mathbf{pa}_V}^{(v')}}$
4     **end**
5 **end**
6 **for** $x \in \mathsf{val}(X)$ **do**
7     Compute $\hat{\psi}(x, \mathbf{c}^{tar}) \triangleq \hat{\mathbb{E}}[Y|do(x), \mathbf{c}^{tar}]$ using $\hat{\mathbb{P}}$ in Equation (2)
8 **end**

**Result:** Return $\hat{\phi}(\mathbf{c}^{tar}) \triangleq \arg\max_x \hat{\psi}(x, \mathbf{c}^{tar})$

---

We call our proposed algorithm Unc_CCB. The training phase of Unc_CCB (given as Algorithm 1a) consists of two phases. In each round of Phase 1 (a fraction $\alpha$ of the rounds), it observes a context from the environment, then uniformly at random chooses an action, and finally observes the remaining variables. The observed values are used to perform standard Bayesian updates over the beliefs about parameters of all relevant CPDs. Since we do not assume any *a priori* beliefs for the agent, this uniform exploration helps the agent enter Phase 2 with reasonable starting beliefs.

In Phase 2 (the remaining $(1-\alpha)$ fraction of the rounds) of training, the agent tries to allocate targeted interventions optimally. To this end, it needs to trade-off between exploring new contexts and gaining more knowledge about already-explored contexts, while taking into account the information leakage resulting from the shared pathways in the causal graph (or equivalently, shared CPDs in the factorized representation of $\mathbb{P}$; Appendix E discusses a related subcase). The novel, entropy-like, Unc measure (discussed below) helps guide the agent in this decision in every round. Algorithm 1b specifies the evaluation phase of Unc_CCB, where the agent is evaluated on simple regret.

**Intuition behind the Unc measure** As mentioned before, performing a targeted intervention given by $(x, \mathbf{c}^{tar})$ would allow the agent to update its beliefs about effects of other targeted interventions $(x', \mathbf{c}^{tar'})$. Intuitively, we would want the algorithm to allocate more samples to those targeted interventions that are most informative about the most "valuable" (in terms of reward $Y$) targeted interventions. Unc captures this intuition by providing a measure of the effect on the agent's knowledge of $\mathbb{E}[Y|do(x'), \mathbf{c}^{tar'}]$ when a targeted intervention $(x, \mathbf{c}^{tar})$ is performed. The agent, then aggregates this over all possible $(x', \mathbf{c}^{tar'})$, providing a measure of overall resulting knowledge from targeted intervention $(x, \mathbf{c}^{tar})$; this helps guide its sample allocation (see Line 11 of Algorithm 1a).

**Definition of the Unc measure** We model the conditional distribution $\mathbb{P}(V|\mathbf{pa}_V)$ for any variable $V$ as a categorical distribution whose parameters are sampled from a Dirichlet distribution. That is, $\mathbb{P}(V|\mathbf{pa}_V) = \mathsf{Cat}(V; b_1, ..., b_r)$, where $(b_1, ..., b_r) \sim \mathsf{Dirc}(\theta_{V|\mathbf{pa}_V})$. Here $\theta_{V|\mathbf{pa}_V}$ is a vector of length, say, $r$. Let denote $\theta_{V|\mathbf{pa}_V}[i]$ denote the $i$'th entry of $\theta_{V|\mathbf{pa}_V}$. We *define* an object called Ent that captures a measure of our knowledge of the CPD:

$$\mathsf{Ent}(\mathbb{P}(V|\mathbf{pa}_V)) \triangleq -\sum_i \left[ \frac{\theta_{V|\mathbf{pa}_V}[i]}{\sum_j \theta_{V|\mathbf{pa}_V}[j]} \ln\left( \frac{\theta_{V|\mathbf{pa}_V}[i]}{\sum_j \theta_{V|\mathbf{pa}_V}[j]} \right) \right]$$

Let the parents of $V$ be $PA_V = (U_1, ..., U_p)$; suppose they take a particular set of values $\mathbf{pa}_V = (u_1, ..., u_p)$. We *define* that a CPD $\mathbb{P}(V|\mathbf{pa}_V)$ is *unaffected* by targeted intervention $(x, \mathbf{c}^{tar})$ if there exists $i \in \{1, ..., p\}$ such that either (1) $U_i$ is $X$ and $u_i \neq x$, or (2) $U_i \in \mathcal{C}^{tar}$ and $\mathbf{c}^{tar}\langle U_i \rangle \neq u_i$. In other words, the CPD $\mathbb{P}(V|\mathbf{pa}_V)$ is unaffected by $(x, \mathbf{c}^{tar})$ if doing this targeted intervention and observing all variables $(x, \mathbf{c}^{tar}, \mathbf{c}^{other})$ does *not* enable us to update beliefs about $\mathbb{P}(V|\mathbf{pa}_V)$ using knowledge of $\mathcal{G}$. Now, define

$$\mathsf{Ent}(\mathbb{P}(V|\mathbf{pa}_V)|x, \mathbf{c}^{tar}) \triangleq \begin{cases} \mathsf{Ent}(\mathbb{P}(V|\mathbf{pa}_V)), \text{if } \mathbb{P}(V|\mathbf{pa}_V) \text{ is unaffected by } (x, \mathbf{c}^{tar}) \\ \mathsf{Ent}^{new}(\mathbb{P}(V|\mathbf{pa}_V)), \text{otherwise} \end{cases}$$

where $\mathsf{Ent}^{new}$ is computed by averaging the resulting Ent values over the possible belief updates the agent might make after performing $(x, \mathbf{c}^{tar})$. That is, $\mathsf{Ent}^{new}(\mathbb{P}(V|\mathbf{pa}_V)) = \frac{1}{r} \sum_i \mathsf{Ent}(\mathsf{Cat}(b'_1, ..., b'_r))$ where $(b'_1, ..., b'_r) \sim \mathsf{Dirc}\left(..., \theta_{V|\mathbf{pa}_V}[i-1], \theta_{V|\mathbf{pa}_V}[i] + 1, \theta_{V|\mathbf{pa}_V}[i+1], ...\right)$.

Letting $\mathbf{c}'$ denote $\mathbf{c}^{tar'} \cup \mathbf{c}^{other'}$, we can now define Unc which captures our knowledge of $\mathbb{E}[Y|do(x'), \mathbf{c}^{tar'}]$ if we perform some other targeted intervention given by $(x, \mathbf{c}^{tar})$:

$$\mathsf{Unc}\left(\mathbb{E}[Y|do(x'), \mathbf{c}^{tar'}]|x, \mathbf{c}^{tar}\right) \triangleq \sum_{\mathbf{c}^{other'} \in \mathsf{val}(\mathcal{C}^{other})} \left[ \sum_{V \in \mathcal{C}^{other}} \mathsf{Ent}(\mathbb{P}(V|\mathbf{c}'\langle PA_V \rangle)|x, \mathbf{c}^{tar}) + \right.$$
$$\left. \mathsf{Ent}(\mathbb{P}(Y|x', \mathbf{c}'\langle PA_Y \rangle)|x, \mathbf{c}^{tar}) \right] \cdot \hat{\mathbb{P}}(\mathbf{c}') \cdot \hat{\mathbb{E}}[Y|\mathbf{c}', do(x')]$$

### 3.2 REGRET BOUND

**Theorem 3.1.** *For any $0 < \delta < 1$, with probability $\geq 1 - \delta$,*

$$Regret \leq 3\mathbb{E}_{\mathbf{pa}_Y, \mathbf{c}^{tar}} \left( \sqrt{\left[ \frac{2}{\frac{\alpha T}{N_X} \mathbb{P}(\mathbf{pa}_Y, \mathbf{c}^{tar}) - \epsilon^T_{X, PA_Y}} \right] \ln\left( \frac{2N_X(N_{\mathcal{C}} + |\mathcal{C}|)}{\delta} \right)} \right)$$

$$+ 3 \sum_{C \in \mathcal{C}^{other}} \mathbb{E}_{\mathbf{pa}_C, \mathbf{c}^{tar}} \left( \sqrt{\left[ \frac{2}{\alpha T \mathbb{P}(\mathbf{pa}_C, \mathbf{c}^{tar}) - \epsilon^T_{PA_C}} \right] \ln\left( \frac{2(N_{\mathcal{C}} + |\mathcal{C}|)}{\delta} \right)} \right) \quad (1)$$

*where*

$$\epsilon_{PA_C}^T = \sqrt{\left[\frac{\alpha T}{2}\right] \ln\left(\frac{N_{PA_C}(N_C + |\mathcal{C}|)}{\delta}\right)}, \ \epsilon_{X,PA_Y}^T = \sqrt{\left[\frac{\alpha T}{2}\right] \ln\left(\frac{N_X N_{PA_Y}(N_C + |\mathcal{C}|)}{\delta}\right)}$$

*Proof.* Due to space constraints, we only provide a proof sketch here. The full proof is provided in Appendix A as part of the Supplementary Text. For readability of the proof, we assume that $Y$ and all variables in $\mathcal{C}$ are binary, whereas $X$ can take on any finite set of values. Under Assumption **(A1)** described in Section 2.1, we can factorize as follows:

$$\mathbb{E}[Y|do(x), \mathbf{c}^{tar}] = \sum_{\mathbf{c}^{other} \in \text{val}(\mathcal{C}^{other})} \left[ \mathbb{P}(Y = 1|x, \mathbf{c}\langle PA_Y \rangle) \prod_{c \in \mathbf{c}^{other}} \mathbb{P}(C = c|\mathbf{c}\langle PA_C \rangle) \right] \quad (2)$$

where $\mathbf{c}$ denotes $\mathbf{c}^{tar} \cup \mathbf{c}^{other}$. The proof first bounds errors of estimates of $\hat{\mathbb{P}}(Y = 1|x, \mathbf{c}\langle PA_Y \rangle)$ and $\hat{\mathbb{P}}(C = c|\mathbf{c}\langle PA_C \rangle)$ with high probability using union bounds, concentration bounds, and the fact that the number of rounds in which the agent observes any set of context values is lower bounded by the number of times it sees it in Phase 1. We also make use of the fact that all variables are observed after every round. We then aggregate these bounds of individual CPD estimates to bound the overall estimate of $\hat{\mathbb{E}}[Y|do(x), \mathbf{c}^{tar}]$, and then finally bound regret by utilizing the fact that the algorithm chooses arms based on the estimated $\hat{\mathbb{E}}$. $\qquad\square$

**Discussion** Theorem 3.1 bounds the regret (with high probability). Note that $T \to \infty \implies Regret \to 0$, which shows that the regret bound has desirable limiting behavior.[2] Further, regret is inversely related to $\sqrt{T}$, similar to other simple-regret algorithms such as in Lattimore et al. (2016). However, other terms that are a function of the space of possible interventions ($N_{\mathcal{C}} N_X$) are different since we are in a *contextual* bandit setting (whereas Lattimore et al. (2016) is non-contextual); more specifically, regret is related proportionally to $\sqrt{N_X N_{\mathcal{C}} \ln(N_X N_{\mathcal{C}})}$; in fact, our regret bound is strictly tighter than this. We prove the bound primarily to provide a theoretical guard on regret; we do not claim improved regret bounds since there is no comparable simple-regret contextual bandit regret bound. We show improved performance over baselines empirically in Section 4.

## 4 EXPERIMENTAL EVALUATION

Since there is no setting studied previously that directly maps to our setting, we adapt existing multi-armed bandit algorithms to make use of causal side-information and targeted interventions in a variety of ways to form a set of baselines against which we compare our algorithm's performance.

The baseline algorithms are the following. Std_TS and Std_UniExp follow the standard bandit interaction protocol during training – observe context, choose action, receive reward. They treat each $\mathbf{c}^{tar}$ as a separate problem and learn an optimal arm for each; they differ in that Std_TS uses Thompson Sampling (Agrawal & Goyal, 2012) for each $\mathbf{c}^{tar}$ problem, whereas Std_UniExp does uniform exploration over $\text{val}(X)$. The other set of algorithms are TargInt_TS, TargInt_UniExp and TargInt_TS_UniExp. These treat the space $\text{val}(X) \times \text{val}(\mathcal{C}^{tar})$ as the set of arms for targeted interventions. TargInt_TS performs Thompson Sampling over this space, and TargInt_UniExp performs uniform exploration over this space. TargInt_TS_UniExp is a variation of TargInt_TS that, with some probability,[3] chooses a random targeted intervention; this infuses a controlled degree of uniform exploration into TargInt_TS. All five algorithms update CPD beliefs after each round. No baseline is strictly stronger than another; their relative performance can vary based on the specific setting. We also compare against a non-causal version of algorithm Std_TS, which we call NonCausal_Std_TS, to provide one non-causal baseline (i.e., which does not utilize the causal side-information).

The evaluation phase for each baseline algorithm is analogous to Algorithm 1b – given a $\mathbf{c}^{tar}$, the action with the highest expected reward, based on beliefs at the end of the training phase, is chosen.

---

[2]There is a technical requirement to keep the denominator in Eq (1) greater than 0; however, it is easy to see that this only requires that $T > constant$, so there is no impact on asymptotic behavior.

[3]We choose this probability to be 0.5 for the plots. There is more discussion on this in Appendix F.

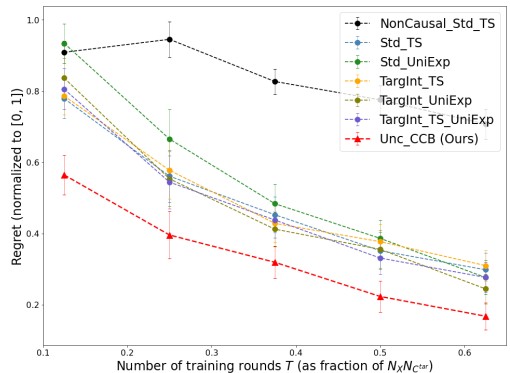
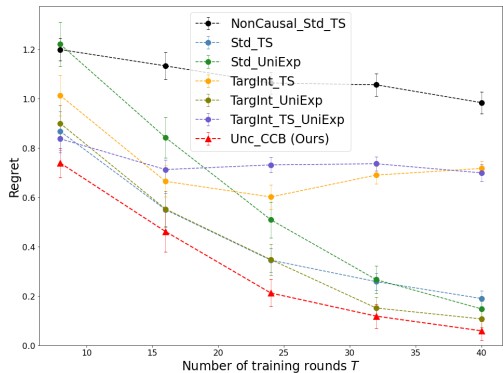

Figure 1: Mean regrets for Experiment 1 (Section 4.1). Regret is normalized to $[0,1]$ and $T$ as fractions of $N_X N_{\mathcal{C}^{tar}}$.

Figure 2: Mean regrets for Experiment 2 (Section 4.1).

## 4.1 PURELY SYNTHETIC DATA

We consider a causal model $\mathcal{M}$ whose causal graph $\mathcal{G}$ consists of the following edges: $C_1 \rightarrow C_0$, $C_0 \rightarrow X$, $C_0 \rightarrow Y$, $X \rightarrow Y$. We assume that $\mathcal{C}^{tar} = \{C_1\}$ and that $\mathcal{C}^{other} = \{C_0\}$. We choose the fraction of Phase 1 rounds $\alpha = 0.5$ as we empirically found that to be a good choice. More details and code are in the Supplementary Material. Additional experiments, analyzing the reason for why our algorithm performs better than baselines, are in Appendix B.

**Experiment 1** Given the wide range of possible settings arising from different parameterizations of $\mathcal{M}$, the first experiment seeks to answer the question: "what is the expected performance of our algorithm over a set of *representative settings*?". Each setting can be interpreted naturally when, for example, we think of $\mathcal{C}^{tar}$ as the set of user features that a recommendation agent observes post-deployment. The settings are chosen to capture the intuition that, typically, the agent sees high-value contexts (i.e., contexts for which, if the agent learns the optimal action, can fetch high rewards) relatively less frequently, say, 20% of the time, but there can be variation in the number of different $\mathcal{C}^{tar}$ *values* over which that probability mass is spread and in how "risky" they are (for example, very low rewards for non-optimal actions). In each run, the agent is presented with a randomly selected setting from this representative set. Results are averaged over 300 independent runs; error bars display $\pm 1.96$ standard errors. For details on specific parameterizations, refer Appendix D.

Figure 1 provides the results of this experiment. The axes are normalized since the representative set of settings could have different ranges for $T$ and for regret. We see that our algorithm performs better than all baselines, especially for lower values of $T$. This, in fact, is a recurring theme – our algorithm's performance improvement is more pronounced in *low T-budget* situations, which, as we stated in Section 1.1, is what we are interested in. Further, Appendix C contains plots for two individual settings from the set, one where Std_TS and Std_UniExp perform significantly better than TargInt_TS, TargInt_UniExp and TargInt_TS_UniExp, and another where it is the converse; in contrast, our algorithm performs close to the best baseline in both. The intuition behind this is that the Unc measure incentivizes exploring new context values (due to its entropy-like nature) while weighting more the contexts that are likely to yield more rewards (based on our current estimates). Also, as expected, NonCausal_Std_TS performs the worst as it ignores the causal graph.

**Experiment 2** To ensure that the results are not biased due to our choice of the representative set, the second experiment asks "what happens to performance when we *directly randomize the parameters* of the CPDs in each run, subject to realistic constraints?". Specifically, in each run, we (1) randomly pick an $i \in \{1, ..., \lfloor N_{C_1}/2 \rfloor\}$, (2) distribute 20% of the probability mass randomly over $C_1$ values $\{1, ..., i\}$, (3) distribute the remaining 80% of the mass over $C_1$ values $\{i+1, ..., N_{C_1}\}$. The $c_1$ values in the 20% bucket are higher-value, while those in the 80% bucket are lower-value. Intuitively, this captures the commonly observed 80-20 pattern (for example, 20% of the users contribute to around 80% of the revenue), but randomizes the other aspects; this gives an estimate of how the algorithms would perform on expectation. The results are averaged over 100 independent runs; error bars display $\pm 1.96$ standard errors. Figure 2 shows that our algorithm performs better than all baselines

in this experiment, with a more pronounced improvement for the smaller values of $T$. For instance, when $T = 24$, our algorithm's mean regret is around $35\%$ lower than that of the next best baseline.

## 4.2 CRM SALES DATA-INSPIRED EXPERIMENT

**Experiment 3** This experiment seeks to answer the question: "how does the algorithm perform on *realistic scenarios*?". We setup the experiment inspired by a real world scenario. Consider a bandit agent that could assist salespeople by learning to decide how many outgoing calls to make in an ongoing deal, given just the type of deal (new business, etc.) and size of customer (small, medium, etc.), so as to maximize a reward metric (which we call the 'expected deal value'). Deal type affects the reward via the source of the lead (for example, chat). The trained agent would be deployed internally or to the company's clients, where it would generally not have access to lead source. The causal graph relating these variables is given in Figure 3a, which was obtained using domain knowledge. 'Deal type' and 'Customer size' are $\mathcal{C}^{tar}$, 'Lead source' is $\mathcal{C}^{other}$, '# outgoing calls' is $X$, and 'Exp. deal value' is $Y$. The parameters of each CPD corresponding to this causal graph was calibrated using proprietary sales data from Freshworks Inc., after some adjustments using domain knowledge (for example, where there were coverage gaps). Note that the algorithms use only the CPDs and not this raw data; in any case, we make the dataset available in anonymized form at `https://doi.org/10.5281/zenodo.5540348` under the *'CC BY-NC 4.0'* license.

Further, we also do a warm-start for all algorithms to account for the fact that, in practice, there is often some starting beliefs about the CPDs from past data, domain knowledge, etc., which can be encoded into the agent's prior beliefs.[4] Specifically, we ran pure random exploration for 15 rounds at the beginning of each algorithm and updated all CPD beliefs; this simulates a warm-start. Due to this, we used $\alpha = 0$ for our algorithm. Results are averaged over 50 independent runs; error bars display $\pm 1.96$ standard errors. Figure 3 shows that our algorithm performs better than all baselines in this realistic setting as well. The Supplementary Material provides the code.

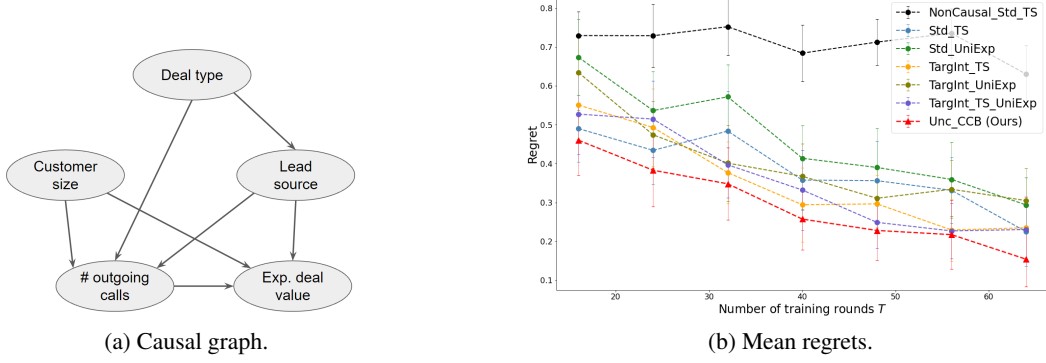

(a) Causal graph.          (b) Mean regrets.

Figure 3: CRM sales data-inspired experiments (Section 4.2)

## 5 CONCLUSION AND FUTURE DIRECTIONS

This work presented a contextual bandits formulation that captures real-world nuances such as the ability to conduct targeted interventions and the presence of causal side-information, along with a novel algorithm that exploits this to achieve improved sample efficiency. In addition to synthetic experiments, we also performed real world-inspired experiments set up using actual CRM sales data.

A useful direction of future investigation is the development of algorithms when $\mathcal{G}$ is unknown or partially known. Another important direction is the development of methods robust to distributional shifts. Distributional shifts have been studied for prediction tasks (Magliacane et al., 2018; Gong et al., 2016) and for causal effects (Bareinboim & Pearl, 2014; Correa & Bareinboim, 2020); it is an interesting question to study this in our setting, when $\mathcal{G}$ remains the same between training and evaluation but $\mathcal{M}$ changes.

---

[4]Other works such as Dimakopoulou et al. (2019) and Liu et al. (2018) have used warm starting as well.

## ACKNOWLEDGEMENTS

This work was partly funded by Freshworks Inc. through a research grant to Balaraman Ravindran and Chandrasekar Subramanian.

## REPRODUCIBILITY STATEMENT

The full proof of Theorem 3.1 is provided in an appendix at the end of this PDF file (after the references) as part of the Supplementary Text. The source code of all the experiments in the main paper, along with a `README` on how to run them, is provided as part of the Supplementary Materials in a file named `Supplemental_Code_Paper1203.zip`. Further, even though the algorithms use only the CPDs and not the raw data, the dataset used to calibrate the real world-inspired experiments (see Section 4.2 of main paper) is made available in anonymized form at `https://doi.org/10.5281/zenodo.5540348` under the *'CC BY-NC 4.0'* license. The other experiments were synthetic and did not involve the use of any dataset.

## ETHICS STATEMENT

This work is a foundational work that provides an improved mathematical framework and algorithm for contextual bandits. Some experiments were set up utilizing data from Freshworks Inc.; the data is released in anonymized form with the consent of the company (link to dataset given in the main paper). To the best of our knowledge, we do not believe there were any ethical issues associated with the development of this work. Further, given the nature of the work as foundational and introducing a new algorithm (and not specific to an application), we do not foresee any specific potential negative ethical issues created by this work. However, we do point out that researchers utilizing this method to their specific applications should adhere to ethical standards of their own (e.g., by avoiding targeting interventions on subpopulations based on racial attributes).

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

## A    APPENDIX (SUPPLEMENTARY TEXT): PROOF OF THEOREM 3.1

This is the full version of the proof sketch presented in Section 3.2 of the paper. For readability of the proof, we assume that $Y$ and all variables in $\mathcal{C}$ are binary, whereas $X$ can have any finite set of values. In Sections A.1 through A.5, we derive the expression for bounding Regret($\mathbf{c}^{tar}$), which is the simple regret *given* $\mathcal{C}^{tar} = \mathbf{c}^{tar}$. Then, in Section A.6, we derive the final expression for bounding overall Regret.

In addition to the notation in Table 1 of the main paper, we use a few other notations to keep the proof more readable:

- $\mathbf{c}$ denotes $\mathbf{c}^{tar} \cup \mathbf{c}^{other}$.
- $\mathbf{a}\langle\mathcal{B}\rangle$ is shortened to $\mathbf{b}$ when it is unambiguous what $\mathbf{a}$ is; for example, $\mathbf{c}\langle PA_V \rangle$ is shortened to $\mathbf{pa}_V$ when it is obvious that $\mathbf{c}$ is the current set of values taken by all the context variables.
- We write $\mathbb{P}(V = v | \mathbf{pa}_V)$ interchangeably with $\mathbb{P}(v | \mathbf{pa}_V)$ since we use small letters (e.g., $v$) to denote the value taken by the random variable denoted by the *respective* capital letter (e.g., $V$).

Now, the factorization under Assumption **(A1)** can be written in this simpler notation as:

$$\mathbb{E}[Y|do(x), \mathbf{c}^{tar}] = \sum_{\mathbf{c}^{other} \in \mathsf{val}(\mathcal{C}^{other})} \mathbb{P}(Y = 1|x, \mathbf{pa}_Y) \prod_{c \in \mathbf{c}^{other}} \mathbb{P}(c|\mathbf{pa}_C)$$

Recollect that we use set operations on vectors as well, wherever there is no ambiguity.

### A.1    BOUNDING THE ERROR OF $\mathbb{P}(Y = 1|x, \mathbf{pa}_Y)$ ESTIMATES

Let $\mathbf{pa}_Y$ denote the value taken by $PA_Y$ (i.e., $Y$'s parents), *excluding* $x$. Say, for a $(x, \mathbf{pa}_Y)$,

$$\mathbb{P}\left[|\hat{\mathbb{P}}(Y = 1|x, \mathbf{pa}_Y) - \mathbb{P}(Y = 1|x, \mathbf{pa}_Y)| \geq \epsilon_{x,\mathbf{pa}_Y}\right] \leq \delta_{x,\mathbf{pa}_Y}$$

Using the union bound, we can rewrite it as

$$\mathbb{P}\left[\forall x, |\hat{\mathbb{P}}(Y = 1|X, \mathbf{pa}_Y) - \mathbb{P}(Y = 1|X, \mathbf{pa}_Y)| \geq \epsilon_{X,\mathbf{pa}_Y}\right] \leq \delta_{X,\mathbf{pa}_Y}$$

In other words, with probability $\geq 1 - \delta_{X,\mathbf{pa}_Y}$, the following event $\mathcal{E}_{X,\mathbf{pa}_Y}$ is true:

$$|\forall x, \hat{\mathbb{P}}(Y = 1|X, \mathbf{pa}_Y) - \mathbb{P}(Y = 1|X, \mathbf{pa}_Y)| \leq \epsilon_{X,\mathbf{pa}_Y}$$

We derive expressions for $\epsilon_{X,\mathbf{pa}_Y}$ and $\delta_{X,\mathbf{pa}_Y}$ in Lemma A.2.

### A.2    BOUNDING THE ERROR OF $\mathbb{P}(C = c|\mathbf{PA}_C)$ ESTIMATES

Now, say, for a $C \in \mathcal{C}$ (taking value $C = c$), and for given $\mathbf{pa}_C$,

$$\mathbb{P}\left[|\hat{\mathbb{P}}(c|\mathbf{pa}_C) - \mathbb{P}(c|\mathbf{pa}_C)| \geq \epsilon_{c|\mathbf{pa}_C}\right] \leq \delta_{c|\mathbf{pa}_C}$$

Note that $\hat{\mathbb{P}}(0|\mathbf{pa}_C) + \hat{\mathbb{P}}(1|\mathbf{pa}_C) = 1$ and $\mathbb{P}(0|\mathbf{pa}_C) + \mathbb{P}(1|\mathbf{pa}_C) = 1$. Now, say the *actual* errors in estimates are given by $\tilde{\epsilon}_c$. That is, $\tilde{\epsilon}_c \triangleq \hat{\mathbb{P}}(c|\mathbf{pa}_C) - \mathbb{P}(c|\mathbf{pa}_C)$. Therefore,

$$\hat{\mathbb{P}}(0|\mathbf{pa}_C) + \hat{\mathbb{P}}(1|\mathbf{pa}_C) = 1 \Longleftrightarrow \mathbb{P}(0|\mathbf{pa}_C) + \tilde{\epsilon}_{0|\mathbf{pa}_C} + \mathbb{P}(1|\mathbf{pa}_C) + \tilde{\epsilon}_{1|\mathbf{pa}_C} = 1$$
$$\Longleftrightarrow 1 + \tilde{\epsilon}_{0|\mathbf{pa}_C} + \tilde{\epsilon}_{1|\mathbf{pa}_C} = 1$$
$$\Longleftrightarrow \tilde{\epsilon}_{0|\mathbf{pa}_C} + \tilde{\epsilon}_{1|\mathbf{pa}_C} = 0$$

Now, this means that

$$|\tilde{\epsilon}_{0|\mathbf{pa}_C}| \leq a \Longleftrightarrow |\tilde{\epsilon}_{1|\mathbf{pa}_C}| \leq a$$

Therefore, the event $|\tilde{\epsilon}_{0|\mathbf{pa}_C}| \leq a$ is *the same event as* the event $|\tilde{\epsilon}_{1|\mathbf{pa}_C}| \leq a$. Therefore, given a $\mathbf{pa}_C$, the following is true:

$$\mathbb{P}\left[\forall c, |\tilde{\epsilon}_{c|\mathbf{pa}_C}| \leq \epsilon_{C|\mathbf{pa}_C}\right] \geq 1 - \delta_{C|\mathbf{pa}_C}$$

which is the same as

$$\mathbb{P}\left[\forall c,\ |\hat{\mathbb{P}}(c|\mathbf{pa}_C) - \mathbb{P}(c|\mathbf{pa}_C)| \le \epsilon_{C|\mathbf{pa}_C}\right] \ge 1 - \delta_{C|\mathbf{pa}_C}$$

where $\delta_{C|\mathbf{pa}_C}$ is a function of $\epsilon_{C|\mathbf{pa}_C}$. Note that we now have $\delta_{C|\mathbf{pa}_C}$ instead of $\delta_{c|\mathbf{pa}_C}$.

In other words, given a $\mathbf{pa}_C$, with probability $\ge 1 - \delta_{C|\mathbf{pa}_C}$, the following event $\mathcal{E}_{C|\mathbf{pa}_C}$ is true:

$$\forall c,\ |\hat{\mathbb{P}}(C = c|\mathbf{pa}_C) - \mathbb{P}(C = c|\mathbf{pa}_C)| \le \epsilon_{C|\mathbf{pa}_C}$$

We derive expressions for $\delta_{C|\mathbf{pa}_C}$ and $\epsilon_{C|\mathbf{pa}_C}$ in Lemma A.1.

## A.3 PUTTING THEM TOGETHER FOR THE OVERALL BOUND FOR REGRET($\mathbf{c}^{tar}$)

Using the union bound, for any $\mathbf{c}^{tar}$, with probability $\ge 1 - \sum_{\mathbf{pa}_Y} \delta_{X,\mathbf{pa}_Y} - \sum_{C \in \mathcal{C}} \sum_{\mathbf{pa}_C} \delta_{C|\mathbf{pa}_C}$, all of $\mathcal{E}_{X,\mathbf{pa}_Y}$ and $\mathcal{E}_{C|\mathbf{pa}_C}$ are true simultaneously $\forall \mathbf{pa}_Y$ and $\forall \mathbf{pa}_C \ \forall C \in (\mathcal{C} \setminus PA_Y)$. Let's call this event $\mathcal{E}$. In other words, under event $\mathcal{E}$, *all* estimates are bounded.

Let $a_{alg} = \hat{\phi}(\mathbf{c}^{tar})$ denote the action chosen by the algorithm in the evaluation phase when presented with $\mathbf{c}^{tar}$; and let $a^*$ denote an optimal action for $\mathbf{c}^{tar}$. Therefore, given $\mathcal{C}^{tar} = \mathbf{c}^{tar}$, under event $\mathcal{E}$,

$$\mathbb{E}[Y|do(a_{alg}), \mathbf{c}^{tar}] = \sum_{\mathbf{c}^{other}} \mathbb{P}(Y = 1|a_{alg}, \mathbf{pa}_Y) \prod_{c \in \mathbf{c}^{other}} \mathbb{P}(c|\mathbf{pa}_C)$$

(Factorization due to $\mathcal{G}$ as earlier)

$$= \sum_{\mathbf{c}^{other}} \mathbb{P}(Y = 1|a_{alg}, \mathbf{pa}_Y) \prod_{c \in \mathbf{c}^{other}} \{\hat{\mathbb{P}}(c|\mathbf{pa}_C) - \tilde{\epsilon}_{C|\mathbf{pa}_C}\}$$

(Since $\tilde{\epsilon}_{C|\mathbf{pa}_C}$ are actual errors.)

$$\ge \sum_{\mathbf{c}^{other}} \{\hat{\mathbb{P}}(Y = 1|a_{alg}, \mathbf{pa}_Y) - \epsilon_{X,\mathbf{pa}_Y}\} \prod_{c \in \mathbf{c}^{other}} \{\hat{\mathbb{P}}(c|\mathbf{pa}_C) - \tilde{\epsilon}_{C|\mathbf{pa}_C}\}$$

(Due to event $\mathcal{E}$)

$$\ge \hat{\mathbb{P}}(Y = 1|do(a_{alg}), \mathbf{c}^{tar}) - \left. \sum_{C \in \mathcal{C}^{other} \cup Y} \sum_{\mathbf{c}^{other}} \mathbb{P}(\mathbf{c}^{other}|\mathbf{c}^{tar})\epsilon_{C|\mathbf{pa}_C} \right|_{x=a_{alg}}$$

(Algebra discussed separately in Section A.3.1)

$$= \hat{\mathbb{E}}[Y|do(a_{alg}), \mathbf{c}^{tar}] - \left. \sum_{C \in \mathcal{C}^{other} \cup Y} \sum_{\mathbf{pa}_C} \mathbb{P}(\mathbf{pa}_C|\mathbf{c}^{tar})\epsilon_{C|\mathbf{pa}_C} \right|_{x=a_{alg}}$$

(Since $Y$ is binary)

$$\ge \hat{\mathbb{E}}[Y|do(a^*), \mathbf{c}^{tar}] - \left. \sum_{C \in \mathcal{C}^{other} \cup Y} \sum_{\mathbf{pa}_C} \mathbb{P}(\mathbf{pa}_C|\mathbf{c}^{tar})\epsilon_{C|\mathbf{pa}_C} \right|_{x=a_{alg}}$$

(Due to proposed algorithm)

Continuing and applying similar steps, we get

$$\mathbb{E}[Y|do(a_{alg}), \mathbf{c}^{tar}] \ge \mathbb{E}[Y|do(a^*), \mathbf{c}^{tar}] - 2 \sum_{\mathbf{pa}_Y} \mathbb{P}(\mathbf{pa}_Y|\mathbf{c}^{tar})\epsilon_{X,\mathbf{pa}_Y}$$

$$- 3 \sum_{C \in \mathcal{C}^{other}} \sum_{\mathbf{pa}_C} \mathbb{P}(\mathbf{pa}_C|\mathbf{c}^{tar})\epsilon_{C|\mathbf{pa}_C}$$

Defining

$$\epsilon'_X \triangleq \sum_{\mathbf{pa}_Y} \mathbb{P}(\mathbf{pa}_Y|\mathbf{c}^{tar})\epsilon_{X,\mathbf{pa}_Y}$$

and

$$\epsilon'_C \triangleq \sum_{\mathbf{pa}_C} \mathbb{P}(\mathbf{pa}_C|\mathbf{c}^{tar})\epsilon_{C|\mathbf{pa}_C}$$

we get that,

$$\mathbb{E}[Y|do(a_{alg}), \mathbf{c}^{tar}] \geq \mathbb{E}[Y|do(a^*), \mathbf{c}^{tar}] - 2\epsilon'_X - 3 \sum_{C \in \mathcal{C}^{other}} \epsilon'_C$$

Therefore, under event $\mathcal{E}$ (that is, with probability $\geq 1 - \sum_{\mathbf{pa}_Y} \delta_{X,\mathbf{pa}_Y} - \sum_{C \in \mathcal{C}} \sum_{\mathbf{pa}_C} \delta_{C|\mathbf{pa}_C}$), for any given $\mathbf{c}^{tar}$,

$$\text{Regret}(\mathbf{c}^{tar}) = \mathbb{E}[Y|do(a^*), \mathbf{c}^{tar}] - \mathbb{E}[Y|do(a_{alg}), \mathbf{c}^{tar}] \leq 2\epsilon'_X + 3 \sum_{C \in \mathcal{C}^{other}} \epsilon'_C \qquad (3)$$

We derive expressions for each of these terms in Sections A.4 and A.5, and then arrive at the final bound in Section A.6.

### A.3.1 ALGEBRA FOR INTERMEDIATE EXPRESSION

We wish to show that

$$\sum_{\mathbf{c}^{other}} \{\hat{\mathbb{P}}(Y = 1|x, \mathbf{pa}_Y) - \epsilon_{X,\mathbf{pa}_Y}\} \prod_{c \in \mathbf{c}^{other}} \{\hat{\mathbb{P}}(c|\mathbf{pa}_C) - \tilde{\epsilon}_{C|\mathbf{pa}_C}\}$$

$$\geq \hat{\mathbb{P}}(Y = 1|do(x), \mathbf{c}^{tar}) - \sum_{C \in \mathcal{C}^{other} \cup Y} \sum_{\mathbf{c}^{other}} \mathbb{P}(\mathbf{c}^{other}|\mathbf{c}^{tar}) \epsilon_{C|\mathbf{pa}_C}$$

Let us denote the expression on the left hand side by $(A)$.

Let $c_i \in \mathbf{c}^{other}$ chosen in a reverse topological order; this ensures that a child is chosen before any of its parents. Let $C_i$ be the corresponding variable in $\mathcal{C}^{other}$. Then we can write

$$(A) = \sum_{\mathbf{c}^{other} \backslash c_i} \left[ \sum_{c_i} \left[ \{\hat{\mathbb{P}}(Y = 1|x, \mathbf{pa}_Y) - \epsilon_{X,\mathbf{pa}_Y}\}\{\hat{\mathbb{P}}(c_i|\mathbf{pa}_{C_i}) - \tilde{\epsilon}_{c_i|\mathbf{pa}_{C_i}}\} \right] \right.$$

$$\left. \prod_{c' \in \mathbf{c}^{other} \backslash c_i} \{\hat{\mathbb{P}}(c'|\mathbf{pa}_{C'}) - \tilde{\epsilon}_{c'|\mathbf{pa}_{C'}}\} \right]$$

$$= \sum_{\mathbf{c}^{other} \backslash c_i} \left[ \sum_{c_i} \left[ \hat{\mathbb{P}}(Y = 1|x, \mathbf{pa}_Y)\hat{\mathbb{P}}(c_i|\mathbf{pa}_{C_i}) - \tilde{\epsilon}_{c_i|\mathbf{pa}_{C_i}}\hat{\mathbb{P}}(Y = 1|x, \mathbf{pa}_Y) \right. \right.$$

$$\left. \left. -\epsilon_{X,\mathbf{pa}_Y}\hat{\mathbb{P}}(c_i|\mathbf{pa}_{C_i}) + \epsilon_{X,\mathbf{pa}_Y}\tilde{\epsilon}_{c_i|\mathbf{pa}_{C_i}} \right] \right] \prod_{c' \in \mathbf{c}^{other} \backslash c_i} \{\hat{\mathbb{P}}(c'|\mathbf{pa}_{C'}) - \tilde{\epsilon}_{c'|\mathbf{pa}_{C'}}\}$$

$$\geq \sum_{\mathbf{c}^{other} \backslash c_i} \left[ \hat{\mathbb{Q}}_{C_i}[Y = 1|x, \mathbf{pa}_{C_i}] - \epsilon_{C_i|\mathbf{pa}_{C_i}} - \hat{\epsilon}_{X,\mathbf{pa}_{C_i}} \right] \prod_{c' \in \mathbf{c}^{other} \backslash c_i} \{\hat{\mathbb{P}}(c'|\mathbf{pa}_{C'}) - \tilde{\epsilon}_{c'|\mathbf{pa}_{C'}}\}$$

where

$$\hat{\mathbb{Q}}_{\mathcal{D}}[Y = 1|x, \mathbf{c}] \triangleq \sum_{\mathbf{d}} \hat{\mathbb{P}}(Y = 1|x, \mathbf{pa}_Y) \prod_{d \in \mathbf{d}} \{\hat{\mathbb{P}}(d|\mathbf{pa}_D, \mathbf{c})\}$$

and

$$\hat{\epsilon}_{X,\mathbf{pa}_{C_i}} \triangleq \sum_{c_i} \epsilon_{X,\mathbf{pa}_Y} \mathbb{P}(c_i|\mathbf{pa}_{C_i})$$

These make use of the fact that $\tilde{\epsilon}_{c_i|\mathbf{pa}_{C_i}} \leq \epsilon_{C_i|\mathbf{pa}_{C_i}}$, $\tilde{\epsilon}_{0|\mathbf{pa}_{C_i}} = -\tilde{\epsilon}_{1|\mathbf{pa}_{C_i}}$ and that $\hat{\mathbb{P}} \leq 1$. We can keep reapplying the above steps in reverse topological order till we exhaust all $c_i$, and we'll get

$$(A) \geq \hat{\mathbb{Q}}_{\mathcal{C}^{other}}[Y = 1|x, \mathbf{c}^{tar}] - \sum_{C \in \mathcal{C}^{other} \cup Y} \sum_{\mathbf{c}^{other}} \mathbb{P}(\mathbf{c}^{other}|\mathbf{c}^{tar}) \epsilon_{C|\mathbf{pa}_C}$$

Noting that $\hat{\mathbb{Q}}_{\mathcal{C}^{other}}[Y = 1|x, \mathbf{c}^{tar}] = \hat{\mathbb{P}}(Y = 1|do(x), \mathbf{c}^{tar})$ gives us the desired inequality.

## A.4 EXPRESSIONS FOR $\delta_{C|\mathbf{pa}_C}$ AND $\epsilon_{C|\mathbf{pa}_C}$

**Lemma A.1.**

$$\mathbb{P}\left[\forall c,\ |\hat{\mathbb{P}}(c|\boldsymbol{pa}_C) - \mathbb{P}(c|\boldsymbol{pa}_C)| \leq \sqrt{\left[\frac{2}{T'\mathbb{P}(\boldsymbol{pa}_C, \boldsymbol{c}^{tar}) - \epsilon_{PA_C}^T}\right] \ln\left(\frac{2}{\delta_{C|\boldsymbol{pa}_C}}\right)}\right] \geq 1 - \delta_{C|\boldsymbol{pa}_C}$$

*Proof.* Let $\mathcal{T}_{\mathbf{pa}_C}$ be the set of time indices during training when $PA_C = \mathbf{pa}_C$ was chosen/seen by the algorithm, and let $T_{\mathbf{pa}_C} = |\mathcal{T}_{\mathbf{pa}_C}|$. Now, for a $C$ that is *observed* and not chosen[5], note that our estimate of $\hat{\mathbb{P}}(C = 1|\mathbf{pa}_C)$ is computed as $(\theta_{C|\mathbf{pa}_C}^{(1)} + 1)/(T_{\mathbf{pa}_C} + 2)$. We approximate[6] this as $\theta_{C|\mathbf{pa}_C}^{(1)}/T_{\mathbf{pa}_C}$. Using Hoeffding's inequality, for any $c$,

$$\mathbb{P}\left[|\hat{\mathbb{P}}(C = c|\mathbf{pa}_C) - \mathbb{P}(C = c|\mathbf{pa}_C)| \geq \epsilon_{C|\mathbf{pa}_C}\right] \leq 2\exp\left(-\frac{T_{\mathbf{pa}_C}}{2}(\epsilon_{C|\mathbf{pa}_C})^2\right)$$

But since already observed, the two events (corresponding to $c = 0$ and $c = 1$) are the same, we have

$$\mathbb{P}\left[\forall c,\ |\hat{\mathbb{P}}(c|\mathbf{pa}_C) - \mathbb{P}(c|\mathbf{pa}_C)| \leq \epsilon_{C|\mathbf{pa}_C}\right] \geq 1 - 2\exp\left(-\frac{T_{\mathbf{pa}_C}}{2}(\epsilon_{C|\mathbf{pa}_C})^2\right)$$

Therefore, we let

$$\delta_{C|\mathbf{pa}_C} \triangleq 2\exp\left(-\frac{T_{\mathbf{pa}_C}}{2}(\epsilon_{C|\mathbf{pa}_C})^2\right)$$

Now, let $T' \triangleq \alpha T$ be the number of rounds of Phase 1. Let $\tilde{T}_{C|\mathbf{pa}_C}$ be the number of times $C$ was updated in Phase 1 as a results of encountering $PA_C = \mathbf{pa}_C$. Note that the mean of $\tilde{T}_{C|\mathbf{pa}_C}$ is $T'\mathbb{P}(\mathbf{c}^{tar})\mathbb{P}(\mathbf{pa}_C|\mathbf{c}^{tar}) = T'\mathbb{P}(\mathbf{pa}_C, \mathbf{c}^{tar})$. By the Hoeffding's inequality,

$$\mathbb{P}\left[\tilde{T}_{C|\mathbf{pa}_C} \geq T'\mathbb{P}(\mathbf{pa}_C, \mathbf{c}^{tar}) - \epsilon_{\mathbf{pa}_C}^T\right] \geq 1 - \exp\left(-\frac{2(\epsilon_{\mathbf{pa}_C}^T)^2}{T'}\right)$$

By the union bound, we have (let's call the event as $\mathcal{E}_{C|PA_C}^T$),

$$\mathbb{P}\left[\forall \mathbf{pa}_C,\ \tilde{T}_{C|\mathbf{pa}_C} \geq T'\mathbb{P}(\mathbf{pa}_C, \mathbf{c}^{tar}) - \epsilon_{PA_C}^T\right] \geq 1 - N_{PA_C}\exp\left(-\frac{2(\epsilon_{PA_C}^T)^2}{T'}\right)$$

where $N_{PA_C} = \prod_{C \in PA_C} N_C$.

Letting $\delta_{C|PA_C}^T \triangleq N_{PA_C}\exp\left(-\frac{2(\epsilon_{PA_C}^T)^2}{T'}\right)$, we have

$$\epsilon_{C|PA_C}^T = \sqrt{\left[\frac{T'}{2}\right]\ln\left(\frac{N_{PA_C}}{\delta_{C|PA_C}^T}\right)}$$

Note that $T_{C|\mathbf{pa}_C} \geq \tilde{T}_{C|\mathbf{pa}_C}$. Therefore, given that $\mathcal{E}_{C|PA_C}^T$ is true, we have

$$\mathbb{P}\left[\forall c,\ |\hat{\mathbb{P}}(c|\mathbf{pa}_C) - \mathbb{P}(c|\mathbf{pa}_C)| \leq \epsilon_{C|\mathbf{pa}_C}\right] \geq 1 - 2\exp\left(-\frac{T_{C|\mathbf{pa}_C}}{2}(\epsilon_{C|\mathbf{pa}_C})^2\right)$$

$$\geq 1 - 2\exp\left(-\frac{T'\mathbb{P}(\mathbf{pa}_C, \mathbf{c}^{tar}) - \epsilon_{C|PA_C}^T}{2}(\epsilon_{C|\mathbf{pa}_C})^2\right)$$

---

[5]This means $C \in \mathcal{C}$ during Phase 1 of the algorithm, or $C \in \mathcal{C}^{other}$ in Phase 2.

[6]It's easy to see that this does not change the asymptotic behavior of the bound.

Defining $\delta_{C|\mathbf{pa}_C}$ as

$$\delta_{C|\mathbf{pa}_C} \triangleq 2 \exp\left(-\frac{T'\mathbb{P}(\mathbf{pa}_C, \mathbf{c}^{tar}) - \epsilon_{C|PA_C}^T}{2}(\epsilon_{C|\mathbf{pa}_C})^2\right)$$

gives us

$$\epsilon_{C|\mathbf{pa}_C} = \sqrt{\left[\frac{2}{T'\mathbb{P}(\mathbf{pa}_C, \mathbf{c}^{tar}) - \epsilon_{C|PA_C}^T}\right] \ln\left(\frac{2}{\delta_{C|\mathbf{pa}_C}}\right)} \tag{4}$$

$\square$

## A.5 Expressions for $\delta_{X,\mathbf{pa}_Y}$, $\epsilon_{X,\mathbf{pa}_Y}$, and $\epsilon'_x$

**Lemma A.2.**

$$\mathbb{P}\left[\forall x, |\hat{\mathbb{P}}(Y = 1|x, \mathbf{pa}_Y) - \mathbb{P}(Y = 1|x, \mathbf{pa}_Y)| \leq \right.$$

$$\left.\sqrt{\left[\frac{2}{\frac{T'}{N_X}\mathbb{P}(\mathbf{pa}_Y, \mathbf{c}^{tar}) - \epsilon_{X,PA_Y}^T}\right] \ln\left(\frac{2N_X}{\delta_{X,\mathbf{pa}_Y}}\right)}\right] \geq 1 - \delta_{X,\mathbf{pa}_Y}$$

*Proof.* As before, let $\mathcal{T}_{x,\mathbf{pa}_Y}$ be the set of time indices during training when $(X, PA_Y) = (x, \mathbf{pa}_Y)$ was chosen/seen by the algorithm, and let $T_{x,\mathbf{pa}_Y} = |\mathcal{T}_{x,\mathbf{pa}_Y}|$. As before, note that our estimate of $\hat{\mathbb{P}}(Y = 1|x, \mathbf{pa}_Y)$ is computed as $(\theta_{Y|x,\mathbf{pa}_Y}^{(1)} + 1)/(T_{x,\mathbf{pa}_Y} + 2)$, which we can approximate as $\theta_{Y|x,\mathbf{pa}_Y}^{(1)}/T_{x,\mathbf{pa}_Y}$. Now, by Hoeffding's inequality

$$\mathbb{P}\left[|\hat{\mathbb{P}}(Y = 1|x, \mathbf{pa}_Y) - \mathbb{P}(Y = 1|x, \mathbf{pa}_Y)| \geq \epsilon_{x,\mathbf{pa}_Y}\right] \leq 2 \exp\left(-\frac{T_{x,\mathbf{pa}_Y}}{2}\epsilon_{x,\mathbf{pa}_Y}^2\right)$$

During Phase 1, let $\tilde{T}_{x,\mathbf{pa}_Y}$ be the random variable denoting the number of times that $(X, PA_Y) = (x, \mathbf{pa}_Y)$ was seen/chosen. Note that, due to the nature of Phase 1 (i.e., each $x$ is chosen with probability $\frac{1}{N_X}$), the mean of $\tilde{T}_{x,\mathbf{pa}_Y}$ is $T'\mathbb{P}(\mathbf{pa}_Y, \mathbf{c}^{tar})\frac{1}{N_X}$. Therefore, using the union bound and the Hoeffding's inequality,

$$\mathbb{P}\left[\forall(x, \mathbf{pa}_Y), \tilde{T}_{x,\mathbf{pa}_Y} \geq T'\mathbb{P}(\mathbf{pa}_Y, \mathbf{c}^{tar})\frac{1}{N_X} - \epsilon_{X,PA_Y}^T\right] \geq 1 - N_X N_{PA_Y} \exp\left(-\frac{2}{T'}(\epsilon_{X,PA_Y}^T)^2\right)$$

Since $\forall(x, \mathbf{pa}_Y)$, $T_{x,\mathbf{pa}_Y} \geq \tilde{T}_{x,\mathbf{pa}_Y}$, we have (let's call this event $\mathcal{E}_{X,PA_Y}^T$),

$$\mathbb{P}\left[\forall(x, \mathbf{pa}_Y), T_{x,\mathbf{pa}_Y} \geq T'\mathbb{P}(\mathbf{pa}_Y, \mathbf{c}^{tar})\frac{1}{N_X} - \epsilon_{X,PA_Y}^T\right] \geq 1 - N_X N_{PA_Y} \exp\left(-\frac{2}{T'}(\epsilon_{X,PA_Y}^T)^2\right)$$

Letting $\delta_{X,PA_Y}^T \triangleq N_X N_{PA_Y} \exp\left(-\frac{2}{T'}(\epsilon_{X,PA_Y}^T)^2\right)$, we have

$$\epsilon_{X,PA_Y}^T = \sqrt{\left[\frac{T'}{2}\right] \ln\left(\frac{N_X N_{PA_Y}}{\delta_{X,PA_Y}^T}\right)}$$

Now, we can get that, for a given $\mathbf{pa}_Y$, $\forall x$, (let's call it event $\mathcal{E}_{X,\mathbf{pa}_Y}$),

$$\mathbb{P}\left[\forall x, |\hat{\mathbb{P}}(Y = 1|x, \mathbf{pa}_Y) - \mathbb{P}(Y = 1|x, \mathbf{pa}_Y)| \leq \epsilon_{X,\mathbf{pa}_Y}\right]$$

$$\geq 1 - \sum_x 2 \exp\left(-\frac{T_{x,\mathbf{pa}_Y}}{2}(\epsilon_{X,\mathbf{pa}_Y})^2\right)$$

$$\geq 1 - 2N_X \exp\left(-\frac{\frac{T'}{N_X}\mathbb{P}(\mathbf{pa}_Y, \mathbf{c}^{tar}) - \epsilon_{X,PA_Y}^T}{2}(\epsilon_{X,\mathbf{pa}_Y})^2\right)$$

Setting

$$\delta_{X,\mathbf{pa}_Y} = 2N_X \exp \left( -\frac{\frac{T'}{N_X}\mathbb{P}(\mathbf{pa}_Y,\mathbf{c}^{tar}) - \epsilon_{X,PA_Y}^T}{2}(\epsilon_{X,\mathbf{pa}_Y})^2 \right)$$

we have,

$$\epsilon_{X,\mathbf{pa}_Y} = \sqrt{\left[ \frac{2}{\frac{T'}{N_X}\mathbb{P}(\mathbf{pa}_Y,\mathbf{c}^{tar}) - \epsilon_{X,PA_Y}^T} \right] \ln \left( \frac{2N_X}{\delta_{X,\mathbf{pa}_Y}} \right)} \qquad (5)$$

$\square$

## A.6 DERIVING THE FINAL REGRET BOUND

The probability that events $\mathcal{E}$, $\mathcal{E}_{PA_C}^T (\forall C \in \mathcal{C})$ *and* $\mathcal{E}_{X,PA_Y}^T$ are all simultaneously true is

$$\geq 1 - \sum_{\mathbf{pa}_Y} \delta_{X,\mathbf{pa}_Y} - \delta_{X,PA_Y}^T - \sum_{C \in \mathcal{C}} \left( \sum_{\mathbf{pa}_C} \delta_{C|\mathbf{pa}_C} + \delta_{PA_C}^T \right)$$

Substituting Equations (4) and (5) back into Equation (3), we get that, with the above probability, for a given $\mathbf{c}^{tar}$,

$$\text{Regret}(\mathbf{c}^{tar}) \leq 3 \left( \sum_{\mathbf{pa}_Y} \mathbb{P}(\mathbf{pa}_Y|\mathbf{c}^{tar}) \sqrt{\left[ \frac{2}{\frac{T'}{N_X}\mathbb{P}(\mathbf{pa}_Y,\mathbf{c}^{tar}) - \epsilon_{X,PA_Y}^T} \right] \ln \left( \frac{2N_X}{\delta_{X,\mathbf{pa}_Y}} \right)} \right.$$

$$\left. + \sum_{C \in \mathcal{C}^{other}} \sum_{\mathbf{pa}_C} \mathbb{P}(\mathbf{pa}_C|\mathbf{c}^{tar}) \sqrt{\left[ \frac{2}{T'\mathbb{P}(\mathbf{pa}_C,\mathbf{c}^{tar}) - \epsilon_{PA_C}^T} \right] \ln \left( \frac{2}{\delta_{C|\mathbf{pa}_C}} \right)} \right)$$

where

$$\epsilon_{C|PA_C}^T = \sqrt{\left[ \frac{T'}{2} \right] \ln \left( \frac{N_{PA_C}}{\delta_{C|PA_C}^T} \right)}$$

and

$$\epsilon_{X,PA_Y}^T = \sqrt{\left[ \frac{T'}{2} \right] \ln \left( \frac{N_X N_{PA_Y}}{\delta_{X,PA_Y}^T} \right)}$$

We set $\tilde{\delta} = \delta_{X,\mathbf{pa}_Y} = \delta_{X,PA_Y}^T = \delta_{C|\mathbf{pa}_C} = \delta_{PA_C}^T, \forall C, \mathbf{pa}_Y, \mathbf{pa}_C$. Thus, for a given $\mathbf{c}^{tar}$, with probability $\geq 1 - N_{PA_Y}\tilde{\delta} - \tilde{\delta} - N_{\mathcal{C}\setminus PA_Y}\tilde{\delta} - |\mathcal{C} \setminus PA_Y|\tilde{\delta}$,

$$Regret(\mathbf{c}^{tar}) \leq 3 \left( \sum_{\mathbf{pa}_Y} \mathbb{P}(\mathbf{pa}_Y|\mathbf{c}^{tar}) \sqrt{\left[ \frac{2}{\frac{T'}{N_X}\mathbb{P}(\mathbf{pa}_Y,\mathbf{c}^{tar}) - \epsilon_{X,PA_Y}^T} \right] \ln \left( \frac{2N_X}{\tilde{\delta}} \right)} \right.$$

$$\left. + \sum_{C \in \mathcal{C}^{other}} \sum_{\mathbf{pa}_C} \mathbb{P}(\mathbf{pa}_C|\mathbf{c}^{tar}) \sqrt{\left[ \frac{2}{T'\mathbb{P}(\mathbf{pa}_C,\mathbf{c}^{tar}) - \epsilon_{C|PA_C}^T} \right] \ln \left( \frac{2}{\tilde{\delta}} \right)} \right)$$

where

$$\epsilon_{C|PA_C}^T = \sqrt{\left[ \frac{T'}{2} \right] \ln \left( \frac{N_{PA_C}}{\tilde{\delta}} \right)}$$

and

$$\epsilon_{X,PA_Y}^T = \sqrt{\left[ \frac{T'}{2} \right] \ln \left( \frac{N_X N_{PA_Y}}{\tilde{\delta}} \right)}$$

Now, noting that $N_{PA_Y}\tilde{\delta} + \tilde{\delta} + N_{C \setminus PA_Y}\tilde{\delta} + |\mathcal{C} \setminus PA_Y|\tilde{\delta} \leq (N_{\mathcal{C}} + |\mathcal{C}|)\tilde{\delta}$, we set $\delta = (N_{\mathcal{C}} + |\mathcal{C}|)\tilde{\delta}$. So we get that with probability $\geq 1 - \delta$,

$$
\text{Regret}(\mathbf{c}^{tar}) \leq 3 \left( \sum_{\mathbf{pa}_Y} \mathbb{P}(\mathbf{pa}_Y|\mathbf{c}^{tar}) \sqrt{\left[ \frac{2}{\frac{\alpha T}{N_X}\mathbb{P}(\mathbf{pa}_Y, \mathbf{c}^{tar}) - \epsilon_{X,PA_Y}^T} \right] \ln\left( \frac{2N_X(N_{\mathcal{C}} + |\mathcal{C}|)}{\delta} \right)} \right.
$$
$$
\left. + \sum_{C \in \mathcal{C}^{other}} \sum_{\mathbf{pa}_C} \mathbb{P}(\mathbf{pa}_C|\mathbf{c}^{tar}) \sqrt{\left[ \frac{2}{\alpha T \mathbb{P}(\mathbf{pa}_C, \mathbf{c}^{tar}) - \epsilon_{PA_C}^T} \right] \ln\left( \frac{2(N_{\mathcal{C}} + |\mathcal{C}|)}{\delta} \right)} \right) \quad (6)
$$

where

$$
\epsilon_{PA_C}^T = \sqrt{\left[ \frac{\alpha T}{2} \right] \ln\left( \frac{N_{PA_C}(N_{\mathcal{C}} + |\mathcal{C}|)}{\delta} \right)}
$$

and

$$
\epsilon_{X,PA_Y}^T = \sqrt{\left[ \frac{\alpha T}{2} \right] \ln\left( \frac{N_X N_{PA_Y}(N_{\mathcal{C}} + |\mathcal{C}|)}{\delta} \right)}
$$

Substituting Equation (6) into the definition of Regret in Section 2 of the main paper, we get the expression in Theorem 3.1.

## B  APPENDIX (SUPPLEMENTARY TEXT): ADDITIONAL EXPERIMENT – UNDERSTANDING THE REASON FOR BETTER PERFORMANCE

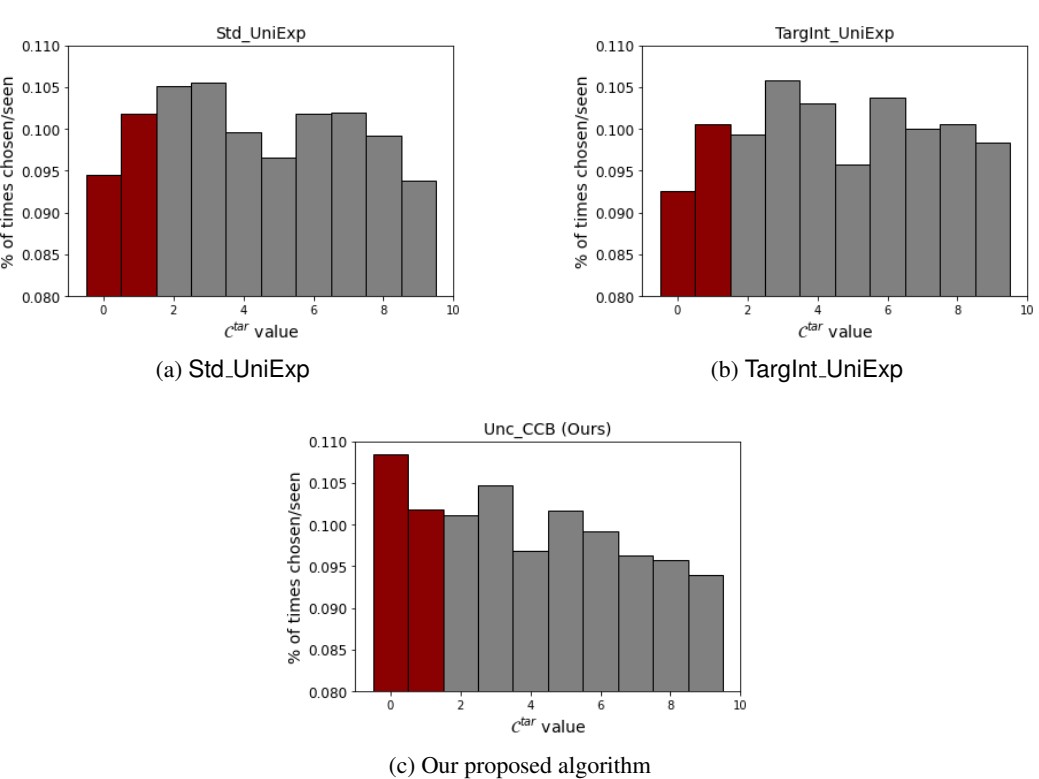

(a) Std_UniExp

(b) TargInt_UniExp

(c) Our proposed algorithm

Figure 4: Frequency of choosing or encountering each value of $\mathcal{C}^{tar}$. Highlighted in red color are the 'high-value' contexts (i.e., contexts for which learning the right actions provides higher expected rewards).

In this analysis, we try to "*understand why* our algorithm exhibits better performance than baselines." We had pointed out intuitively that the Unc measure trades off between exploring new contexts and

learning more about explored contexts based on (a) current knowledge of the various context-action pairs, and (b) current estimates of the 'value' (i.e., rewards obtainable) of context-action pairs. This section attempts to zoom into this behavior.

Specifically, we consider one of the settings (setting #1) from the representative set used in Experiment 1; it has $N_{\mathcal{C}^{tar}} = 10$, all equally likely, but $\mathcal{C}^{tar} \in \{0, 1\}$ are more valuable than others (i.e., learning the right actions for these contexts can give higher rewards); see code (config_setting1.py) for more details. We zoom into the case when $T = 15$. We do 500 independent runs, count the number of times every possible $\mathcal{C}^{tar}$ value is encountered or chosen, and plot the frequencies. We compare our algorithm's behavior with two representative baselines – Algorithm Std_UniExp (which does standard contextual bandit interactions) and Algorithm TargInt_UniExp (which does targeted interventions); the effect is similar for the other two baselines as well. As seen in Figure 4, our algorithm chooses the higher value contexts ($\mathcal{C}^{tar} \in \{0, 1\}$) with relatively higher frequency than the baselines, while still ensuring good exploration of other contexts – in line with the intuition discussed in the main paper.

## C  APPENDIX (SUPPLEMENTARY TEXT): MORE DETAILS ABOUT EXPERIMENT 1

In the discussion about Experiment 1 in the main paper, it was mentioned that Appendix C contains plots for two individual settings from the representative set – one where Algorithms Std_TS and Std_UniExp perform significantly better than Algorithms TargInt_TS, TargInt_UniExp and TargInt_TS_UniExp, and another where it is the converse – while our algorithm performs close to the best baseline in both. Figure 5 presents those plots. Each result is based on 50 runs.

## D  APPENDIX (SUPPLEMENTARY TEXT): REGARDING PARAMETERIZATIONS USED IN THE EXPERIMENTS

For the specific parameterizations of all settings used in all the experiments, refer to the README file in Supplemental_Code_Paper1203.zip as part of the Supplemental Material.

## E  APPENDIX (SUPPLEMENTARY TEXT): MORE DISCUSSION ON THE CASE WHERE $\mathcal{C}^{other} = \varnothing$

If $\mathcal{C}^{other} = \varnothing$ (that is, if it is empty), it might appear as if there would not be any information leakage. However, this is not true, and information leakage can still exist. For instance, consider the same graph as in used in Section 4.1. But now suppose that $\mathcal{C}^{tar} = \{C_1, C_0\}$, whereas $\mathcal{C}^{other} = \varnothing$. Consider two targeted interventions, $(x, c_0, c_1)$ and $(x, c_0, c_1')$, where $c_1 \neq c_1'$. Note that the distribution of $Y$ after these two targeted interventions would, respectively, be

$$\mathbb{P}(Y|do(x), (c_0, c_1)) = \mathbb{P}(Y|x, c_0)$$
$$\mathbb{P}(Y|do(x), (c_0, c_1')) = \mathbb{P}(Y|x, c_0)$$

The common term $\mathbb{P}(Y|x, c_0)$ enables information leakage between these two targeted interventions. To see this, note that if the agent conducts targeted intervention $(x, c_0, c_1)$, then it is able to update its beliefs about $\mathbb{P}(Y|x, c_0)$ which improves its estimates of $(x, c_0, c_1')$ as well due to the shared term. So $\mathcal{C}^{other} = \varnothing$ does not preclude information leakage.

Instead, suppose that we are considering the interventions $(x, c_0, c_1)$ and $(x, c_0', c_1)$, where $c_0 \neq c_0'$. In this case, information leakage gains will not be possible. However, our algorithm would still be able to achieve better performance compared to baselines by *balancing* between exploring new $(x, \mathbf{c}^{tar})$ pairs and learning more about already explored "valuable" $(x, \mathbf{c}^{tar})$ pairs. To see this, note that in the definition of Unc in Section 3.1 the "information" component is weighted by the "value" component (the last two terms, namely, $\hat{\mathbb{P}}(\mathbf{c}') \cdot \hat{\mathbb{E}}[Y|\mathbf{c}', do(x')]$).

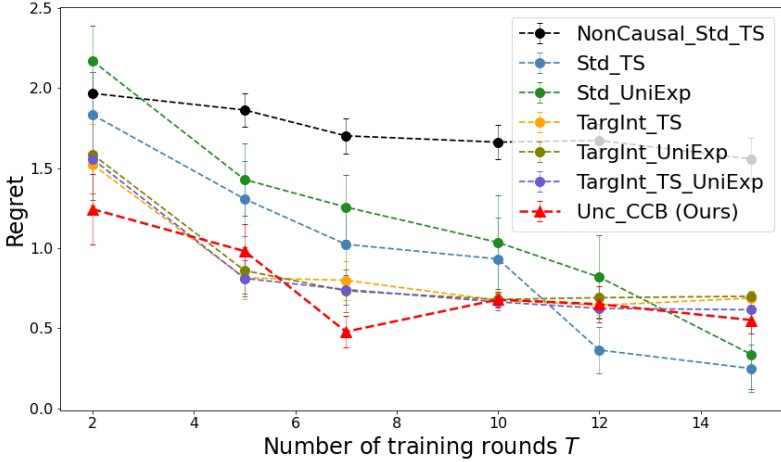

(a) Algorithms TargInt_TS, TargInt_UniExp and TargInt_TS_UniExp perform better than other baselines.

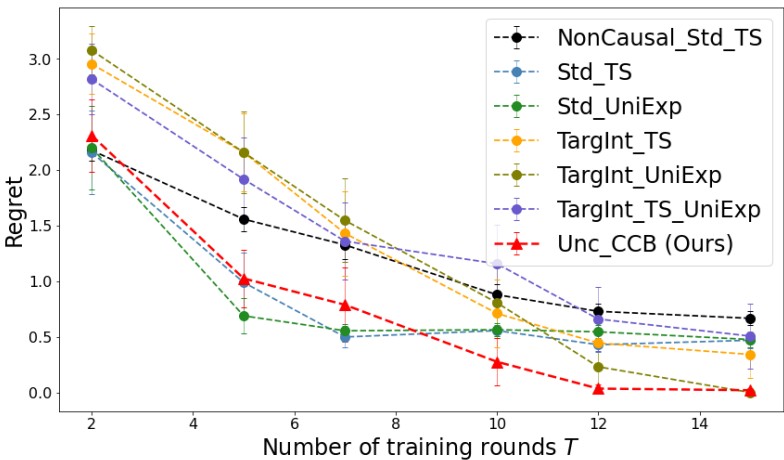

(b) Algorithms Std_TS, Std_UniExp perform better than other baselines.

Figure 5: Different baselines outperform each other in two different settings; however, our algorithm performs close to the best baseline in both.

## F APPENDIX (SUPPLEMENTARY TEXT): REGARDING THE CHOICE OF THE PROBABILITY OF UNIFORM EXPLORATION IN TARGINT_TS_UNIEXP

Thompson Sampling aims to minimize cumulative regret in a best-arm identification setting (Agrawal & Goyal, 2012). For cumulative regret minimization, it performs better than simple exploration algorithms such as $\epsilon$-greedy, by controlling the exploration based on the agent's current beliefs. However, as discussed in the main part of the paper, in our setting, the agent is faced with the task of learning a *policy*, while balancing exploration of new contexts and learning more about already explored contexts. Thus, apart from having standard Thompson Sampling as a baseline (TargInt_TS), we also create a variation of this (called TargInt_TS_UniExp) where, in each round, the algorithm, with some probability $\lambda$, chooses a targeted intervention uniformly at random (from the space $\mathsf{val}(X) \times \mathsf{val}(\mathcal{C}^{tar})$) for exploration; with probability $1 - \lambda$, it chooses the targeted intervention given by Thompson Sampling. This probability $\lambda$ allows us to directly control the degree of *uniform* exploration.

In Experiment 1 (Section 4.1), we chose the probability of uniform exploration $\lambda = 0.5$ for TargInt_TS_UniExp. This was to act as a midpoint between the TargInt_TS algorithm (attempting to optimize cumulative regret) and TargInt_UniExp (which does uniform exploration). We noted

that our algorithm continues to perform better than all baselines. Figure 6 shows that this continues to be true even when $\lambda$ in TargInt_TS_UniExp is not as large as 0.5, by setting $\lambda = 0.2$.

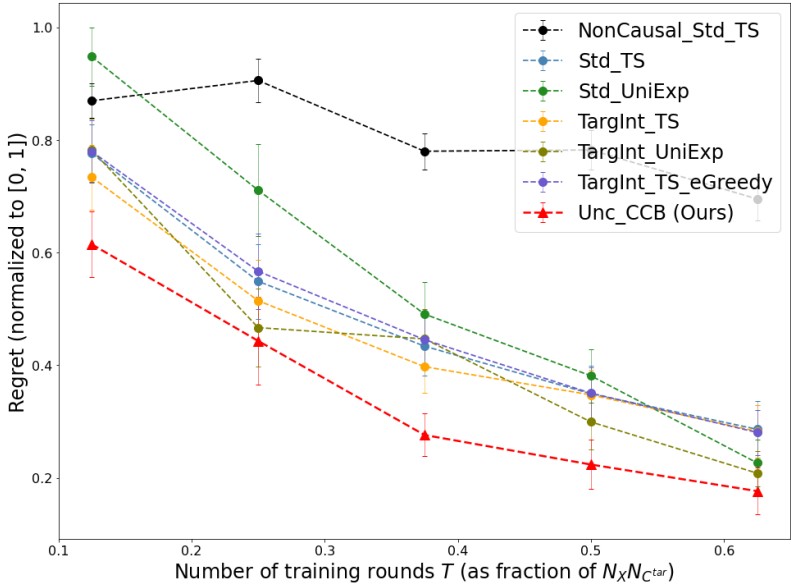

Figure 6: Experiment 1 results when probability of uniform exploration in TargInt_TS_UniExp was chosen to be 0.2 (instead of 0.5). Our algorithm continues to perform better than all baselines.

