# OpenReview forum: "Causal Contextual Bandits with Targeted Interventions"
_ICLR.cc/2022/Conference — ICLR 2022 Poster_

### Official Review · Reviewer_DesA · 2021-10-31

**Correctness:** 3
**Technical Novelty And Significance:** 2
**Empirical Novelty And Significance:** 2
**Recommendation:** 5
**Confidence:** 3

**Main Review:**

Thanks for bringing an interesting bandit setup and proposing an intuitive solution.

First, the assumption on the causal graph seems less accurately described.
Currently, the model the authors are considering seems a Markovian model where there exists no unmeasured confounders. However, it is less clear since, the authors are well-aware of what unobserved confounders are. Then, the assumption (A1) says that any variable that confounds a target context C and reward Y is also in the target contexts C where a sufficient condition for (A1) is C-target being ancestral.
However, In Equation 2, the authors factorized P(bold c_other | bold c_target) as the product of P(c_other | Pa_c_other ), which is not always plausible if you allow unobserved confounders (applying chain rule and conditional independence to prune conditionals). This error indicates that the authors are not quite careful/sure about imposing assumptions on the underlying causal graphs.
(Yet, it is true that one can estimate P(bold_c_other | bold_c_target) as a whole. )

Second, experimental setup is rather naive. In the training phase, there is no reason for an agent to pursue Standard Thompson Sampling (TS) since the task here is not to minimize cumulative regret. The focus here is pure exploration. Similarly, other baseline TargInt_TS is not a proper exploratory agent. As you can see TarInt UniExp performs even better than TargInt_TS. In other words, TargInt UniExp should be considered the minimal baseline, and one may figure out what can be better baselines to compete against the proposed method. For example, what if we use the upper confidence bounds instead of the current estimate multiplied by entropy. Or randomly sample E[Y|d,do(x’)] from posterior distribution instead of using hat E[Y|d,do(x’)]. There are many reasonable possibilities. How about simply dropping uncertainty ? What is the contribution of this entropy term? What if Gini-index? and so on.

Hence, the authors may focus on clarifying the assumptions and analyze the method thoroughly comparing to other possible stronger baselines.


**Summary Of The Paper:**

This paper proposes a contextual bandit problem where an agent is capable of intervening on a targeted population (e.g., a user whose characteristic matches a specific context) instead of getting a context stochastically sampled from the underlying distribution over contexts. The authors proposed an intuitive solution that a targeted intervention should be performed for a specific context with an accompanying action such that, after the intervention, the uncertainty weighted by currently estimated rewards goes down most.


**Summary Of The Review:**

The paper has a certain interesting problem with an intuitive solution. Yet, the theoretical and empirical analysis seem a bit lacking. Further, the assumptions need to be further clarified.

---

### Official Review · Reviewer_swHk · 2021-11-02

**Correctness:** 3
**Technical Novelty And Significance:** 2
**Empirical Novelty And Significance:** 2
**Recommendation:** 5
**Confidence:** 3

**Main Review:**

Strengths
1. The paper provides a complete analysis of a reasonable theoretical guarantee on regret.
2. The simulation and real-data experiments also show good performance of the proposed method, compared with some other methods considering contextual variables.

Weaknesses
1. I wonder what’s the comparison (in simulation or theoretically) between the proposed method and some existing method that does not account for contextual variables (use the same policy for all observed environments)?
My intuition is that the latter would still perform fairly well when contextual variables have values with high probability mass. Would one outperform another in different cases of the underlying distribution for (X, Y, C), or the proposed one would always be better?
2. In section 2.1, the authors claimed that the ordinary interactive bandit setting is “robust to unobserved confounders”. Does this claim hold when the proposed framework uses targeted intervention where the values of contextual variables are restricted?

**Summary Of The Paper:**

This paper considers the problem of interactive decision-making given some context variables from the environment, which is referred to as contextual bandit setting in the paper. The proposed method is novel in the sense that (a) the analyst can choose to make an intervention on a targeted subpopulation (b) it incorporates causal structure among the context variables and the intervention variable.


**Summary Of The Review:**

Overall I think it is an interesting problem to account for context variables in interactive decision making. The proposed method is intuitive and the authors provide theoretical and numerical evidence for its performance. Still, I would be more convinced if there is a direct comparison between the proposed method with an existing method that does not account for context variables.

---

### Official Review · Reviewer_i8oB · 2021-11-02

**Correctness:** 3
**Technical Novelty And Significance:** 3
**Empirical Novelty And Significance:** 3
**Recommendation:** 6
**Confidence:** 3

**Main Review:**

I have reviewed a previous version of this paper. Most of my comments regarding clarity in that version are now resolved: in particular, the notation in Section 2 is more consistent, and the baselines for the experiments are introduced more clearly.

My main concern was that the paper does not particularly explain well how does the side information obtained from the causal structure lead to more efficient exploration. I appreciate the addition of the paragraph "Intuition behind the Unc measure" in the new version, which addresses this concern. It is mentioned there that an intervention $(x,\mathbf{c}^\mathrm{tar})$ could reveal information about other interventions $(x',\mathbf{d}^\mathrm{tar})$ and that $\mathrm{Unc}$ is a measure of this potential information gain. A concrete example of how this happens in one of the settings considered during the experiments would be even more helpful.

However, what still remains unclear to me is the role of auxiliary context variables $\mathcal{C}^\mathrm{other}$ in this story. I might be mistaken about this but the problem seems to be interesting only when $\mathcal{C}\neq \mathcal{C}^\mathrm{tar}$ and $\mathcal{C}^\mathrm{other}\neq\emptyset$. Otherwise, it could be treated as a regular contextual bandit problem, where $\mathcal{C}\cap \mathrm{PA}_Y$ is treated as the context variables (letting $\bar{\mathcal{C}}=\mathcal{C}\cap \mathrm{PA}_Y$, no intervention $(x,\bar{\mathbf{c}})$ would be informative about other interventions on $X\cup\bar{\mathcal{C}}$). If this is the case, I would have like to see a discussion of it. For instance, in what real world problems do we have $\mathcal{C}\neq \mathcal{C}^\mathrm{tar}$? If this is not the case, then how does Unc_CCB perform compared with standard contextual bandit algorithms (that only consider $\bar{\mathcal{C}}$ as the context and ignore other variables) when $\mathcal{C}= \mathcal{C}^\mathrm{tar}$? If there is still an improvement, then what is the source of the gain there?


**Summary Of The Paper:**

This paper formulates a contextual bandit problem with causal side-information, where the agent has the option to perform targeted interventions. Then, the paper proposes an algorithm for solving this problem and shows that its regret is bounded.

**Summary Of The Review:**

The paper is clearer compared with an older version I have reviewed but I still have questions regarding how the knowledge of a causal structure helps with learning.

---

### Official Review · Reviewer_cUNV · 2021-11-06

**Correctness:** 4
**Technical Novelty And Significance:** 3
**Empirical Novelty And Significance:** 2
**Recommendation:** 6
**Confidence:** 3

**Main Review:**

### Strengths:

1. The paper is well-explained, well-organized and paints a linear, coherent story (albeit the notation is a bit difficult to follow). The paragraph on discussion of the theorem is useful and brief paragraphs explaining the intuition behind Unc measure are helpful for the reader.
2. The paper explores a novel formulation of contextual bandits motivated by a real-world software agent application and lays the foundation for approaching this setting by proposing a solution approach.
3. The paper also presents theoretical analysis of the proposed algorithm. (I haven’t checked the theorem math and details, so would defer to other reviewers to check for sanity here)

### Weaknesses:

1. It is not discussed why the two-phase algorithm mentioned is optimal. For example, instead of splitting into blocks of $\alpha$ and $1-\alpha$ could it have been better to keep oscillating between uniform interventions and targeted interventions during training? Or could there be some other way of allocating interventions that is optimal or is there some reasoning for the proposed method that I’m missing?
2. In the experimental section, each of the baselines considered has a specific, significant shortcoming making it appear somewhat like a strawman. I’m wondering whether it could have been possible to compare the proposed algorithm with other possible ways of leveraging targeted interventions to make the comparison more competitive?
3. The notation is difficult to follow. In section 2, what is $\mathcal{D}$? I thought the same was denoted by $\mathcal{C}$ earlier? In the same paragraph, $\mathcal{C}^{tar}$ is defined but $\textbf{c}^{tar}$ is used in some places.
4. Terms like belief are mentioned in the introduction but are only explained later in Table 1. I’m not sure Table 1 clearly defines ‘belief’ either — is it a distribution over possible parameter values? That makes it hard to follow the algorithm and parse terms like updateBelief().
5. In a broader picture of the paper, I’m confused how the casual side-information piece relates to the targeted interventions? It seems like these are two disconnected ideas/features of the problem setting that the paper tries to capitalize on. Does the key idea of the paper in any way rely on leveraging both these phenomena simultaneously rather than independently?


**Summary Of The Paper:**

The paper studies a contextual bandit setting with two unique features: (1) the learning agent has ability to perform targeted interventions during the learning phase (ability to select target sub-populations or context) and (2) it also has access to and integrates casual information in the setting. The key motivation is that this setting captures real-world scenarios such as software product experimentation. One key contribution of the work is a new algorithm that achieves better sample efficiency. The algorithm is evaluated using synthetic experiments and real-world-inspired synthetic experiments.


**Summary Of The Review:**

The paper studies a new contextual bandit setting motivated from real-world scenarios and proposes a new algorithm and presents theoretical guarantees on the same. There is experimental evaluation via simulations but I’m unsure if the baselines considered are competitive. The paper is well-written generally.

---

### Decision · Program_Chairs · 2022-01-20

**Decision:**

Accept (Poster)

**Comment:**

This paper considers a new setting of contextual bandits where the learning agent has the ability to perform interventions on targeted subsets of the population. The problem is motivated from software product experimentation but with more general applicability. The paper provides a method under this setting, with both empirical and theoretical support. Reviewers agree that this is an interesting setting, and the paper contributes new results. The initial concerns on assumptions, correctness and experiments were addressed in the rebuttal. I thus recommend accept. The authors should include the response carefully in the final version.